# Sharp increase in inequality in education in times of the COVID-19-pandemic

Carla Haelermans[1,2]*, Roxanne Korthals[1,3,4], Madelon Jacobs[1], Suzanne de Leeuw[1,4], Stan Vermeulen[1,4], Lynn van Vugt[1], Bas Aarts[1,4], Tijana Prokic-Breuer[1,3,4], Rolf van der Velden[1,2,4], Sanne van Wetten[1,4], Inge de Wolf[1,3,4]

1 Research Centre for Education and the Labour Market (ROA), School of Business and Economics, Maastricht University, Maastricht, The Netherlands, 2 Initiative for Education Research (NRO), Den Haag, The Netherlands, 3 Inspectorate of Education, Utrecht, The Netherlands, 4 Education Lab, School of Business and Economics, Maastricht University, Maastricht, The Netherlands

* carla.haelermans@maastrichtuniversity.nl

**Data Availability Statement:** Data cannot be shared publicly because they are part of the data at Statistics Netherlands and cannot be exported from

## Abstract

The COVID-19-pandemic forced many countries to close schools abruptly in the spring of 2020. These school closures and the subsequent period of distance learning has led to concerns about increasing inequality in education, as children from lower-educated and poorer families have less access to (additional) resources at home. This study analyzes differences in declines in learning gains in primary education in the Netherlands for reading, spelling and math, using rich data on standardized test scores and register data on student and parental background for almost 300,000 unique students. The results show large inequalities in the learning loss based on parental education and parental income, on top of already existing inequalities. The results call for a national focus on interventions specifically targeting vulnerable students.

## Introduction

The COVID-19-pandemic of early 2020 interrupted or even completely halted the learning of children in many countries around the world. Globally, schools were closed for an average of almost 95 school days between March 2020 and February 2021 [1], which is equivalent to almost half a school year in countries where a school year is 40 weeks. In many western countries, schools continued to teach remotely. However, there were many challenges related to distance learning, such as access to digital learning devices and digital learning gaps [e.g., 2–4]. This prompted serious worries of social-emotional problems and learning loss. Despite the lack of adequate data in many countries, some studies appeared on the use of online learning tools by students [e.g., 5] and on the effect of distance schooling on performance and learning gains of students in primary education. Although some studies did not find significant learning losses [e.g., no effects on reading in the USA [6], no learning deficits on schools with a large share of students with advantaged backgrounds in Australia [7]], most studies report negative consequences of the school closures for children's educational development [Belgium [2], UK [4, 8], Italy [9], Switzerland [10], Germany [11], USA [12–15], Norway [16]]. For

the secured virtual environment at Statistics Netherlands. However, all data underlying the results presented are available at Statistics Netherlands and use can be requested by national and international researchers via the Netherlands Cohort Study on Education (NCO) and Statistics Netherlands; data access requests may be sent to info@nationaalcohortonderzoek.nl. An extensive description of the data and the access procedure can be found in Haelermans, C., T. Huijgen, M. Jacobs, M. Levels, R. van der Velden, L. van Vugt and S. van Wetten (2020). Using Data to Advance Educational Research, Policy and Practice: Design, Content and Research Potential of the Netherlands Cohort Study on Education. European Sociological Review, 36(4), 643-662, https://academic.oup.com/esr/article/36/4/643/5871552?login=true.

**Funding:** The authors gratefully acknowledge financial support from The Netherlands Organisation for Health Research and Development (ZonMw, https://www.zonmw.nl/nl/) (project 10430 03201 0014). The funders had no role in study design, data collection and analysis, decision to publish, or preparation of the manuscript. Grant acquired by CH, RK, TP-B, RvdV, IdW.

**Competing interests:** The authors have declared that no competing interests exist.

higher education, the results are less consistent: some find negative effects [17] while others indicate that distance learning might have made students more efficient [18] or see little effects [19].

There are worries that some groups of students experienced lower learning gains due to the school closures and the COVID-19-pandemic than others. Our hypothesis is that the school closures and the pandemic resulted in increased inequality in skill development for students from specific backgrounds (socio-economic status, income and migration background). There is reason to believe that inequalities have indeed increased due to the school closures. For instance, in the Netherlands especially lower-educated parents felt less capable in helping their children with their schoolwork [20, 21]. In the United Kingdom, we see that middle class parents spent more time on home schooling than parents from the working class [22, 23]. If this is the case, and these learning losses persist, they can be detrimental for development of skills in the long run, and in turn lead to an increase of the existing inequalities in opportunities in education and on the labor market [24].

Previous studies on inequalities based on socioeconomic background variables in students' learning gains during the school closures, were hampered by data limitations. Some were limited by their data on educational performance: they used relatively small samples, focused on a specific region rather than a national representative sample, or were limited to only one grade level or subject [6, 8, 9, 12, 13, 16]. Others had limited information on students' background characteristics. They used school level indicators [2, 7, 11] or relatively uninformative categories. For example, a recent study based on Dutch data was only able to distinguish between families in which at least one parent had a lower secondary degree (92%) and families in which both parents had less than a lower secondary degree (8%) [3]. Our study improves upon these studies for several reasons: 1) as a result of the widespread use of standardized testing in the Netherlands, we have a large sample of students who were tested shortly before and after the first lockdown, 2) we have rich student background information at the individual level, including multiple student background variables that indicate whether a student is disadvantaged or not, based on meaningful and informative categories and 3) we focus on effects for separate grade levels, and three different subjects (reading, spelling and mathematics) showing large variation, instead of only looking at overall effects or one subject.

Therefore, in the study at hand, we are able to look in greater detail at background differences between students and present results showing that the learning loss due to the school closures are unequally distributed: students from disadvantaged backgrounds have suffered much more than their fellow students. To show this, we use standardized test score data from the Netherlands and link this to register data on student and parental background for primary school students.

## COVID-19 educational policy changes in the Netherlands

Although compulsory education starts at age 5, Dutch children generally enter primary school at age 4. They remain in primary school up to age 12, after which they enter secondary school and are tracked according to their ability. Almost all schools in the Netherlands are public schools (99%) funded by the Ministry of Education, Culture and Science [25].

February 27, 2020 the first COVID-19 patient was reported in the Netherlands. Primary schools closed at March 16, 2020 and reopened May 11, 2020. Vulnerable children, and children of parents with essential occupations who could not work from home, were allowed to come to school during the school closure. However, these children usually followed the same program as the children who had to stay at home and comprised only around 5% of all children in this first period of school closure. Up to June 7, 2020 children only went to school half

of the time. In this way, groups were smaller and it was easier to keep distance. From June 8 onwards schools went back their usual schedule. Children and teachers were still urged to stay at home when they showed any symptoms associated with COVID-19.

The Netherlands was relatively well equipped for online education, as a total of 96% of the Dutch households have internet access at home [26]. Additionally, the Dutch government made 2.5 million euros available in March to support online learning. This money was used to buy laptops and/or to provide internet access for 7,000 students. This money was supplemented with another 3.8 million euros in May 2020. In total, over 16,000 laptops and tablets were financed in this way [27]. Nevertheless, the school closure happened relatively sudden with no time to prepare. Teachers had to improvise, students suddenly had to structure their own school day, and parents had to act as teachers for their children. Although we do not know exactly how much education children received while schools were closed, there are strong indications that children spent less time on their education than usual. Studies in Germany and Switzerland report considerable reduction in studying time during school closings [28, 29]. Moreover, a survey among Dutch parents revealed that parents, especially in disadvantaged families, did often not feel equipped to support their children during the school closing [20].

Children in countries with longer school closings and less internet access might have experienced larger learning losses and larger inequalities because they experienced prolonged periods of limited and unequal excess to education. In line with this hypothesis, a recent study in Italy [9] reports larger learning losses (0.19 SD) than previous studies in the Netherlands (0.08 SD) [3]. In Italy schools were closed for 15 weeks (one of the first and longest school closings in Europe). Moreover Italy has one of the lowest share of households with a broadband connection [30] and 12% of the students between 6 and 17 years old did not have access to a computer or digital tools at home in 2018/2019 [31]. However, contradicting the idea that longer school closings result in larger learning deficits, a study in Belgium—where schools closed for 8.5 weeks—reports a reduction in mathematics scores of 0.19 SD [2] which is similar in size to the effects found in Italy (15 weeks). Altogether, more research based on country comparisons is needed to be able to state that longer lockdowns result in larger learning deficits and an increase in educational inequalities.

## Materials

In the Netherlands, students take standardized tests throughout grades 1 to grade 6 in primary education. These standardized tests come from different suppliers, with the largest supplier being CITO, with which we collaborated for this paper. Furthermore, schools use administration systems to store the information about the standardized test scores. Three administration systems exported the data on standardized test scores from school year 2013/2014 onwards as part of the Netherlands Cohort Study on Education (NCO) project, a national project initiated by the Dutch Research Council (for a description of this project, see [32]). With permission of the schools, the administration system exports the data on the standardized test scores to Statistics Netherlands, who pseudonominize the student-id and school-id. Before any data was exported, parents were informed about the project and data export by the school, and were given the opportunity (during 4 to 6 weeks) to object against export of their child(ren)'s data (by informing the school written or orally). The school registered any objections in their administration system, and data was not exported from those students whose parents objected.

The data was collected over a period of three months with two exports from the administration systems, the first export took place on the 30th of November 2020, the second on the 18th

**Table 1. Collection of data on standardized tests per school year and grade.**

|  | Grade 1 | Grade 2 | Grade 3 | Grade 4 | Grade 5 |
|---|---|---|---|---|---|
| **School year 2013/2014** |  |  |  |  |  |
| **School year 2014/2015** |  |  |  |  |  |
| **School year 2015/2016** |  |  |  |  |  |
| **School year 2016/2017** |  |  |  |  |  |
| **School year 2017/2018** |  |  |  |  |  |
| **School year 2018/2019** |  |  |  |  |  |
| **School year 2019/2020** |  |  |  |  |  |

of January 2021. In this export, information is collected from school years 2013/2014 to 2019/2020 and gradually consists of more and more students in more and more grades. For more information, see Table 1. In total, 1,319 schools and unique information of 291,635 students was gathered on standardized test scores. After cleaning the data, the total sample used for analyses of this paper comes down to 201,819 students in 1,178 schools.

## National standardized test scores

From grade 1 to grade 6, students take standardized tests twice a year, a midterm test, most often administered to students in the months January and February of the school year, and an end-of-term test, mostly administered in the months June and July, right before the summer holidays. For most schools, these are digital tests. Some schools opt for the pen-and-paper version. Due to the school closure in the spring of 2020, for the school year 2019/2020 the end-of-term test could be postponed until after the summer holidays, which many schools did: about a quarter of schools decided to test their students after the summer holidays in August, September or even October. Test supplier CITO made a recalculation for the test scores in August, September and October to account for the extra time until the test, and make them comparable to the test scores of students who made the test before summer. Note that the tests written during the pandemic were exactly the same type and format as before the pandemic, and there is no within school variation between the type and format of the tests before and during the pandemic.

We use test scores in the domains reading, spelling and math. Table 2 shows the number of test records and unique students per domain. The test in math contains both abstract problems and contextual problems that describe a concrete task. The reading test assesses the student's ability to understand written texts, including both factual and literary content. Lastly, the test in spelling asks students to write down a series of words (no verbs), demonstrating that they have learned the spelling rules. For reading, there is no mid-term test in the first grade, therefore the learning gains between the midterm test and end-of-term test cannot be calculated for grade 1.

The learning gains are defined based on the standardized test scores and are calculated by subtracting the score on the midterm test from the end-of-term test of each domain within a

**Table 2. Number of test records per domain per grade.**

|  | Grade 1 | Grade 2 | Grade 3 | Grade 4 | Grade 5 | Total |
|---|---|---|---|---|---|---|
| **Reading** | n/a | 42,120 | 56,746 | 58,872 | 44,081 | 201,819 |
| **Spelling** | 54,424 | 52,166 | 59,558 | 60,762 | 42,864 | 269,774 |
| **Math** | 61,144 | 55,801 | 62,578 | 68,709 | 63,207 | 311,439 |

school year, with the condition that the student must have taken a midterm and end-of-term test within the same school year at the same school. To remove the influence of outliers, the top and bottom 1% of the absolute learning gains scores are not included in the analyses.

## Student background variables

In the secured virtual environment of Statistics Netherlands, standardized test scores can be matched to background information of the students and their parents. Note that the data in the environment of Statistics Netherlands are pseudonymized such that data are fully anonymous to the researchers that use these data. The data on background information that we use are the highest education level and highest income of parents, student migration background and student gender. Parental education is defined as low when the highest obtained degree of (one) the parents is in pre-vocational secondary education (vmbo b/k), or a degree in upper secondary vocational education (mbo 1), or grades 7 to 9 in pre-vocational secondary education (vmbo gl/tl) or senior general secondary education or university preparatory education (1), middle when a degree in upper secondary vocational education level 2, 3 or 4, or when completed senior general secondary education or university preparatory education (2), and high when a degree at a university of applied sciences is attained or higher (3). This division of parental education over three categories is also being used in the Netherlands Cohort Study on Education and leads to a division in categories that is not only relevant at the content level, but also provides us with large enough groups to have statistical power. Highest parental income is defined as low when the highest income of one of the parents is below the minimum income level (1), middle when the income is higher than minimal level but below twice the minimum income level (2) and high when the income of one of the parents is higher than twice the minimum income. Students' migration background is defined as either having a Dutch background or a western background, or a non-western background. Students with a Dutch or western background are combined into one category because the data contains only very few students with a western background, and the results of these two groups are very comparable. In terms of parental education and household income, students in our sample with a non-western migration background are more likely to come from households with relatively low educated parents (26% compared to 6% for the native Dutch and western migrant student sample) and a relatively low income (45% compared to 16%). Lastly, the gender of the student is defined as male or female.

## Representativeness

The data on standardized test scores are only available for schools who gave permission to export the test scores from their administrative system to Statistics Netherlands. As a result, we do not have full population data and consequently selectivity of the sample might play a role. In the schoolyear 2019/2020, we had a total number of 6,174 primary schools in the Netherlands. The 1,178 schools in our sample therefore comprise a proportion of 19% of the total number of schools. Two main sources of selectivity into the sample can be identified. First, the schools that decided to participate in the data collection might not be random. In exchange for sharing the standardized test score, schools received a report on the performance of their school relative to other schools with a comparable student population. We can expect that active schools, which are keen to monitor their progress, are especially interested in the reports and more likely to participate in the data collection project. Second, not every student is tested. Schools tend to exclude students who are absent (e.g., due to illness) or have a very large learning loss. For these students, schools feel a test is not possible or useful. Usually, the number of students per school which are excluded from the standardized tests is relatively small. However

in 2020, after the school closed for several weeks, more schools decided to skip the standardized tests for a larger share of the student population. It is reasonable to assume that students with larger learning losses are less often tested. Therefore, it is likely that our data is not representative for the whole population, and additional tests on our sample in comparison to the full population confirm this. Table 3 shows the representativeness of our sample in comparison to the full population (based on the National Cohort Study on Education; [in Dutch

**Table 3. Representativeness of sample compared to full population on student and school background variables.**

| | Full population | Sample |
|---|---|---|
| Variables | Percentage | Percentage |
| *Gender* | | |
| Female | 49.32 | 49.75 |
| *Migration background* | | |
| Dutch & western migration background | 82.22 | 75.95 |
| Non-western migration background | 17.51 | 24.04 |
| Missing | 0.27 | 0.01 |
| *Parental income* | | |
| Low income | 21.20 | 24.06 |
| Medium income | 53.38 | 50.54 |
| High income | 24.02 | 24.53 |
| Missing | 1.41 | 0.87 |
| *Parental education* | | |
| Low educated | 10.06 | 11.50 |
| Medium educated | 29.87 | 29.03 |
| High educated | 47.59 | 48.95 |
| Missing | 12.48 | 10.52 |
| *School size* | | |
| Less than 141 students | 36.71 | 28.95 |
| Between 141–220 students | 30.27 | 31.26 |
| More than 220 students | 33.01 | 39.79 |
| *School level pct of low educated parents* | | |
| Below 5,5% | 33.12 | 32.22 |
| Between 5,5% and 12% | 33.47 | 32.78 |
| Above 12% | 33.41 | 35.00 |
| *Urbanisation level* | | |
| Low (< 500 adresses/km2) | 7.86 | 5.63 |
| Limited (500–1000 adresses/km2) | 21.87 | 14.41 |
| Medium (1000–1500 adresses/km2 | 17.58 | 12.50 |
| Strong (1500–2500 adresses/km2) | 30.93 | 32.66 |
| Very strong (> = 2500 adresses/km2) | 21.77 | 34.81 |
| *Denomination* | | |
| Public school | 29.79 | 30.55 |
| Schools based on philosophies | 6.10 | 3.77 |
| Schools based on religious beliefs | 64.02 | 65.69 |
| *Observations* | | |
| Total number of students | 2,458,376 | 263,553 |
| Total number of schools in 2019/2020 | 6174 | 1178 |

Note: N of students is based on the number of unique students in the years 2017/2018, 2018/2019 and 2019/2020.

abbreviated as NCO] [32] on student and school background characteristics. Overall, we see that our sample is over-represented in students with a non-western migration background, and students with low parental income. Furthermore, schools in our sample tend to be larger schools located in more urbanized areas.

To limit the impact of selectivity and over-representation of certain students and schools, we use inverse probability weights. In calculating the weights, we use population data on all students enrolled in Dutch primary education and calculate the probability to be in our test score dataset separately per academic year, grade, and test subject domain as a function of students' observable characteristics. These characteristics are parental education, income, migration background, gender, percentage of students with low educated parents at the school, number of students at the school, urbanisation level (based on location of the school), province (based on location of the school) and school denomination.

## Descriptive statistics

Table 4 shows the unstandardized learning growth for the three domains for the 2 years before the pandemic and the year of the pandemic separately. It also shows the learning growth split by group of parental education. Table 4 is used to calculate the normal average learning growth per week in the 20 weeks between midterm and end-of-term test, and the deviation from this in the COVID-19-year. For example, if the normal learning growth for reading is 7 (in 20 weeks time), and during the pandemic it's only 5, the decline in learning growth in weeks is $(20-((20/7)*5) = 5.7$.

## Methods

In order to estimate the effect of the COVID-19 related school closure on students' learning gain, we compare the learning gain between the midterm and the end-of-term test of the COVID-19-exposed cohort (2019/2020) to the learning gain of students from the two previous cohorts using OLS regressions. To account for potential differences in observable characteristics between students of different cohorts, we add controls for student gender, student household income, migration background, and parental educational background. Further, since for some students of the 2019/2020 cohort the end-of-term test was postponed until the start of the next academic year, we add a dummy indicating whether the test was taken at the end of the 2019/2020 academic year or at the beginning of 2020/2021, resulting in the following regression equation, resembling a difference-in-differences design:

$$\Delta y_{ij} = \alpha + T_{ij}\beta + X_{ij}\gamma + \varepsilon_{ijs} \tag{1}$$

Where $\Delta y_{ij}$ stands for the difference in achievement between the end-of-term test and the midterm for student $i$ in grade $j$. $T_{ij}$ is an indicator for the COVID-19 exposed 2019/2020

**Table 4. Descriptive statistics unstandardized learning gains per domain.**

| | | Reading | Spelling | Math |
|---|---|---|---|---|
| **Overall** | Learning growth 2017&2018 | 6.99 | 25.59 | 15.52 |
| | Learning growth 2019 | 5.17 | 22.08 | 13.50 |
| **Low-parental education** | Learning growth 2017&2018 | 6.05 | 25.44 | 15.56 |
| | Learning growth 2019 | 4.27 | 20.28 | 12.31 |
| **Middle-parental education** | Learning growth 2017&2018 | 6.53 | 25.97 | 15.75 |
| | Learning growth 2019 | 4.21 | 21.42 | 13.14 |
| **High-parental education** | Learning growth 2017&2018 | 7.53 | 25.85 | 15.52 |
| | Learning growth 2019 | 5.94 | 22.99 | 14.00 |

cohort, $X_{ij}$ is a vector consisting of the aforementioned control variables, and $\varepsilon_{ijs}$ is the school-level clustered error term. $\beta$ is our coefficient of interest, which captures the difference in average learning gain between the COVID-19 exposed 2019/2020 cohort and the average learning gain of the (pooled) preceding two cohorts (2017/2018 and 2018/2019).

Identification of the COVID-19 effect hinges on the assumption that the learning gain of the different cohorts would have followed a similar trend in the absence of the pandemic. While this assumption is fundamentally untestable, we can provide supporting evidence for it by looking at the variability of learning gains for all grades over time. If these trends are stable, we can be reasonably sure that the difference between the 2019/2020 cohort and the previous two cohorts was caused by the impact of the pandemic. The results of these analyses can be found in Figs 5–10.

In order to estimate the heterogeneous impact of the COVID-19-pandemic along student background characteristics, we add an interaction between the treatment-dummy and the student characteristic of interest to the regression. This results in the following equation:

$$\Delta y_{ij} = \alpha + T_{ij}\beta_0 + T_{ij}C_{ij}\beta_1 + X_{ij}\gamma + \varepsilon_{ijs} \tag{2}$$

Where $C_{ij}$ stands for one of the aforementioned student characteristics: gender, parental education, household income, and migration background. The vector of control variables $X_{ij}$ still includes all other student characteristics. As a robustness check, we also present results of analyses where apart from the interaction we do not include any of the other control variables, with similar results (see Tables 5–10). Finally, a concern could be raised that some of the student characteristics we observe are capturing similar things. For example, parental education and household income are likely to be strongly correlated. In order to isolate the additional impact of COVID-19 along (for example) household income, we therefore run analyses where we control for the interaction between parental education and the treatment-dummy in addition to the interaction with household income. In addition to household income, we do this for student gender and migration background as well, resulting in the following equation:

$$\Delta y_{ij} = \alpha + T_{ij}\beta_0 + T_{ij}C_{ij}\beta_1 + + T_{ij}E_{ij}\beta_2 + X_{ij}\gamma + \varepsilon_{ijs} \tag{3}$$

With $E_{ij}$ standing for the highest level of obtained parental education. As mentioned before, in our main specification we use inverse probability weighting to obtain results representative for the whole Dutch primary school population. As a robustness check, we run the same analyses without employing weights as well as using entropy-balancing weights ensuring covariate balance between the COVID-19 exposed cohort and the control cohorts (similar to the method used by [3]), and obtain similar results (see Tables 5–10).

## Results

In this section, we show the consequences for inequality during the COVID-19-pandemic by comparing the learning gains of students in pre-COVID-19 times (school years 2017/2018 and 2018/2019) to the learning gains since the COVID-19-pandemic (school year 2019/2020) by estimating Eqs (1) through (3). The results are presented in Figs 1–4. Fig 1 presents the results of Eq (1), estimating the overall impact of COVID-19 on student learning gains. Fig 2 shows the result of Eq (2), estimating the disparate impact along students' parental education. Figs 3 and 4 shows the results from Eq (3), estimating the disparate impact along students' household income and migration background, in addition to the differences along parental education. Tables 11–15 show all the underlying regression results.

Fig 1 shows that in all domains, students have a lower learning gains during the COVID-19-pandemic compared to the growth of students in previous cohorts. On average there is 0.14

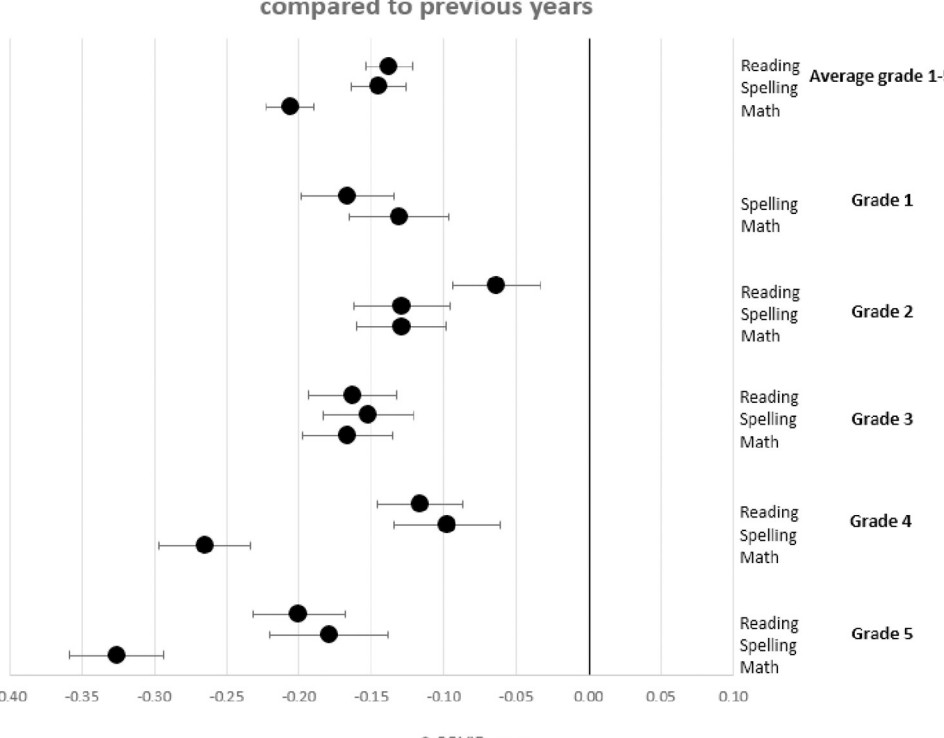

**Fig 1. Standardized coefficients learning gains in COVID-19-year as compared to previous years (zero line).**

standard deviations (SD) learning loss in reading, 0.15 SD in spelling and 0.21 SD in math. Looking at the different grades, we see a gradual increase in the learning loss from grade 1 to grade 5 onwards across all domains, with some outliers, like for instance spelling in grade 4. For reading, we see students experience about 0.06 to 0.20 SD learning loss compared to students from previous cohorts. Looking at spelling shows a similar result, where students experience about 0.13 to 0.18 SD learning loss. Math shows the largest deficits in learning with on average 0.13 (grade 1) to 0.33 (grade 5) SD learning loss.

Although most students learned less in 2019/2020 than their peers in previous cohorts, some students show larger learning loss than others, leading to (increasing) inequality between students. We look at four dimensions of inequality: (1) by parental education, (2) by family income, (3) by migration background and (4) by gender.

Fig 2 shows that children with low-educated parents learned less between the midterm and end-of-year test than their peers with high-educated parents, and that the differences are largest in grades 1, 2 and 3, and for spelling and math(note that alternative specifications in which we use four categories of parental education, or in which we use three categories which are not based on parental education, but on the indication (used for funding purposes) whether a child is a regular child, has a disadvantaged background or a very disadvantaged background, yield very similar results and the same conclusions.) The results show, for example, that children of high-educated parents experience about 0.1 SD more learning gains during the year 2019/2020 compared to children of low-educated parents. In other words, the learning loss due to school closings is larger for students of low-educated parents, and inequalities have grown because of this. The differences between students of high- and low-educated parents are

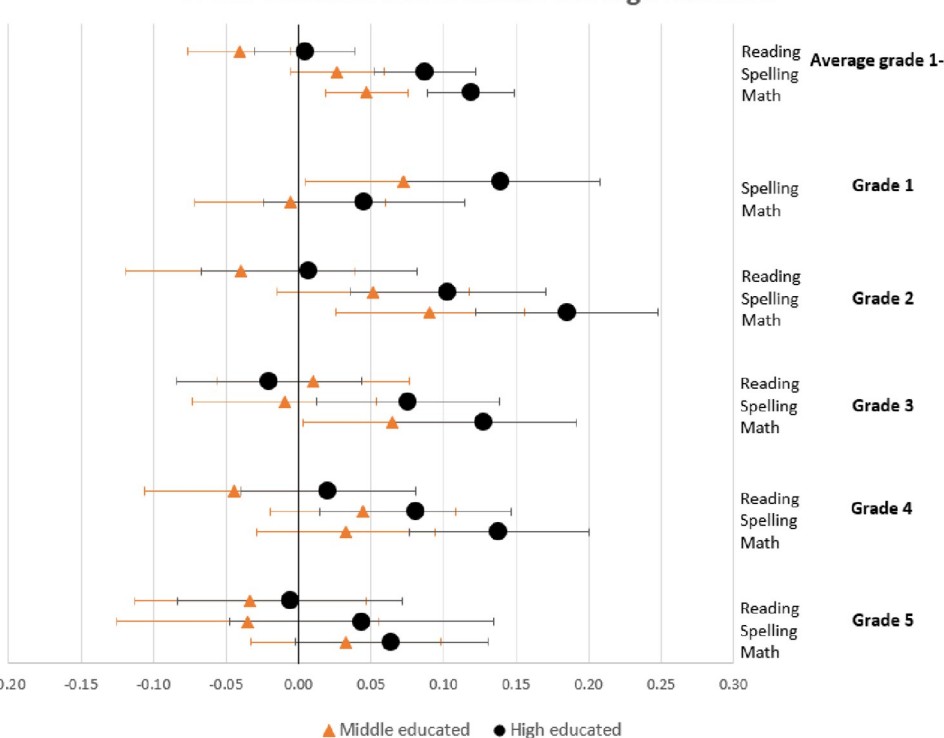

**Fig 2.** Standardized coefficients of learning gains in COVID-19-year 2019/2020 of low-educated (zero line) versus middle- and high-educated.

statistically significant for spelling and math but not for reading, implying that the role of parental background on educational development during the first school closure due to the COVID-19-pandemic is largest for math and spelling, and less pronounced for reading comprehension. Altogether, these findings show that the existing differences in learning gains based on parental education prior to the COVID-19-pandemic have increased during the spring of 2020 when learning was disrupted. These increased differences based on parental education are not surprising, since students were more dependent on the help their parents could provide with schoolwork during the school closure. This finding is also confirmed by other studies: parents in the Netherlands with lower educational attainment felt less capable to help their children with schoolwork [20, 21].

Parental income also plays a role: Fig 3 shows that children from medium and higher income households increase their scores between the midterm and end-of-year test more strongly in the COVID-19-year than their peers from a family with a lower household income, with the largest effects in grades 2 and 3, and for spelling and math. For example, children from medium and high household income experience about 0.05 SD more learning gains than children in low-income households. Note that the relation between income and learning gains is additive to the additional role of parental education during the pandemic. We explicitly take into account the effect of parental education on learning gains and the additional role during the pandemic, and we still see an effect of household income during the school closure. However, it is not surprising that we find an effect of parental income on top of the effect of parental education: parents with higher household income were more likely to afford additional

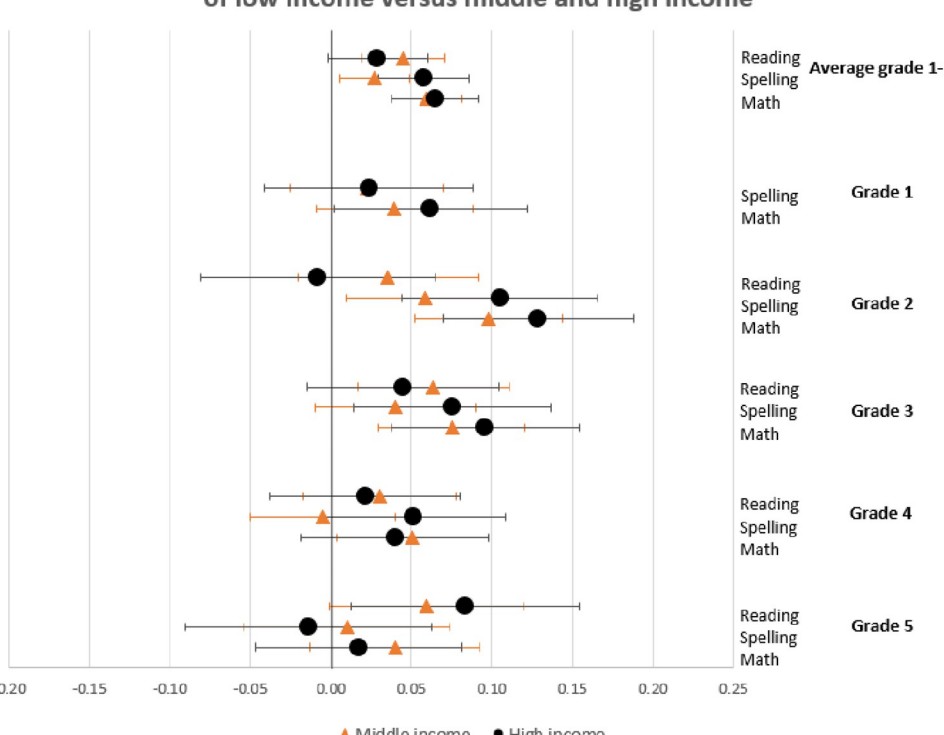

**Fig 3. Standardized coefficients of learning gains in COVID-19-year 2019/2020 of low-income (zero line) versus middle- and high-income.**

help for their children during their time at home. One study suggested that they provide more private access to additional online learning materials [33].

We also looked at the role of migration background, again on top of effects of parental education. The results in Fig 4 indicate that, conditional on the effect of parental education, overall, students with a non-western migration background did not perform significantly worse than other students (native and with a western migration background) during the COVID-19-pandemic. We only find a small significant result for math for grade 2 and 3, and for all grades taken together. Note that, if we do not condition on effects of parental education, we do find significant differences for migration background. Hence, overall, we find that the increased inequality during the pandemic is based on parental education and parental income rather than on migration background.

Lastly, we find that there are no significant differences for gender, neither with nor without controlling for the effect of parental education. Girls seem to perform slightly worse on reading and math but the coefficients are small and almost only statistically significant when all grade levels are taken together (see Table 14).

## Robustness checks

Our preferred model, used throughout the main text, includes inverse probability weights to obtain results that are representative for the entire Dutch primary school population, and we compare the learning gains between the midterm and the end-of-term test in the COVID-19-year to the gains in the two years prior. Furthermore, in the analyses mapping the disparate

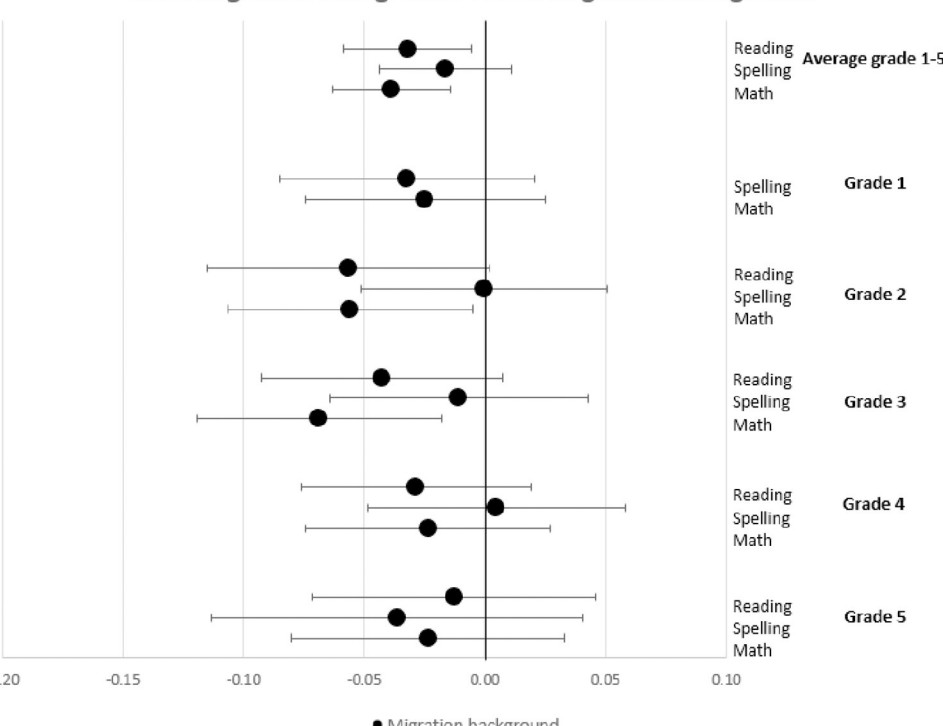

**Fig 4. Standardized coefficients of learning gains in COVID-19-year 2019/2020 of no non-western migration background (zero line) versus non-western migration background.**

impact of COVID-19 along several student characteristics we control for all other background characteristics. These choices could potentially influence our results and their interpretation. Therefore, in this section we show the results of additional analyses where we change the specification of our main model.

To demonstrate how the choice of including inverse probability weights influences the results, Tables 5 and 6 show the results when using entropy-balancing weights and unweighted regressions, respectively. While there are some slight differences in terms of significance levels of certain coefficients, the overall pattern of lower learning gains during the COVID-19-year for students from low-income households and low educated parents, especially in spelling and math, remains similar in magnitude.

In Table 7 we run our main specification without controlling for student gender and migration background, and the dummy accounting for whether the end-of-term test was taken at the end of the school year of 2019/2020 or at the beginning of the 2020/2021 school year. It could be that the interaction effects found on student household income and parental educational background only hold conditional on these other student characteristics. If so, this complicates the interpretation of our results. Fortunately, the exclusion of these additional control variables does not change the found associations.

Table 8 shows how the results change when we control for students' learning gains that they obtained in the previous year. Including prior performance helps in addressing potential differences between cohorts in the trend of their cognitive development. However, the downside of this specification is that we do not observe prior performance for students that are in the first grade (or second grade for the reading domain), and they drop from the analyses as a

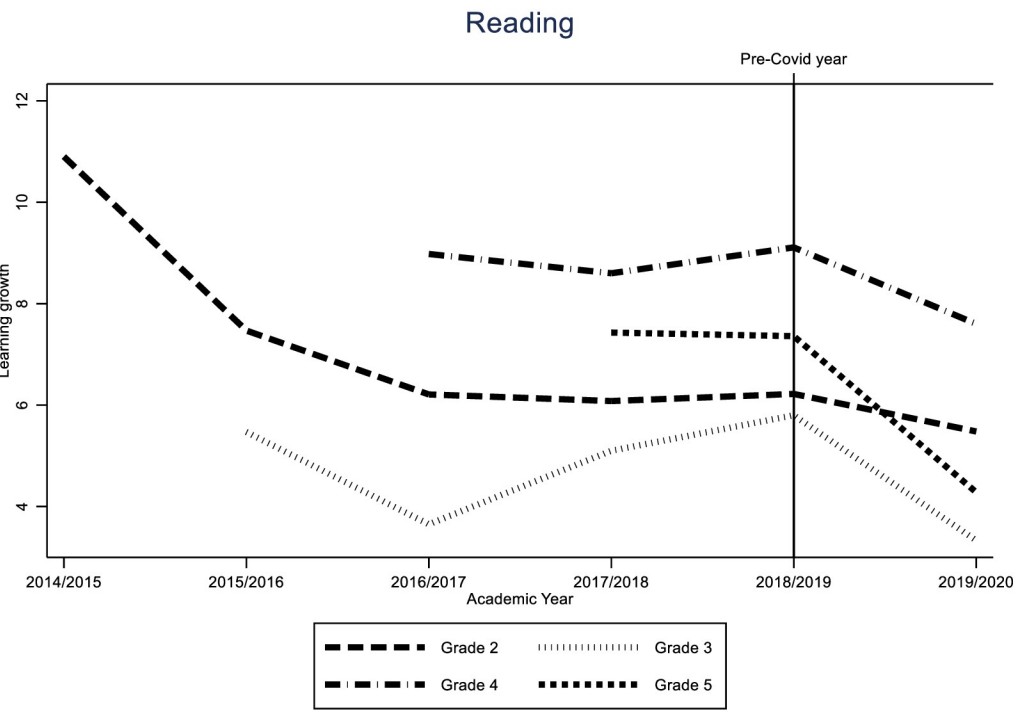

**Fig 5. Trends in learning gain over time–reading.**

result. For the other grades, the results are similar to the main specification. Prior learning gains are significantly positively related to later learning gains for all but one subgroup (grade 5 spelling), but its inclusion does not alter the size and significance of the main results.

Table 9 adds school fixed effects to the regression. With this we control for the possibility that time-invariant factors at the school level are driving our results. This could be the case, for example, when students are strongly sorted into schools according to their background characteristics. In this case the association between student characteristics and learning gains could be driven by (unobserved) differences between schools that house different kinds of student populations. The results from Table 9 however show that including school fixed effects does not change the patterns of the found associations.

Finally, Table 10 shows the results of our main specification as well as the previously discussed robustness checks for the pooled sample over all grades. This table further demonstrates that while there are some slight differences in terms of significance between specifications when looking at grades separately, the overall picture of the disparate impact of COVID-19 on student learning gains along household income and parental education levels remains strong and is robust to various alternative model specifications.

## Trends in learning gain over time

The interpretation of the differences in learning gains between the cohort affected by the COVID-19 induced lockdown and previous cohorts as attributable to the impact of COVID-19 hinges on the assumption that learning gains would have been similar in the absence of the pandemic. While this assumption is untestable, we can provide supporting evidence for it by looking at the variability of learning gains for all grades over time. If these trends are relatively

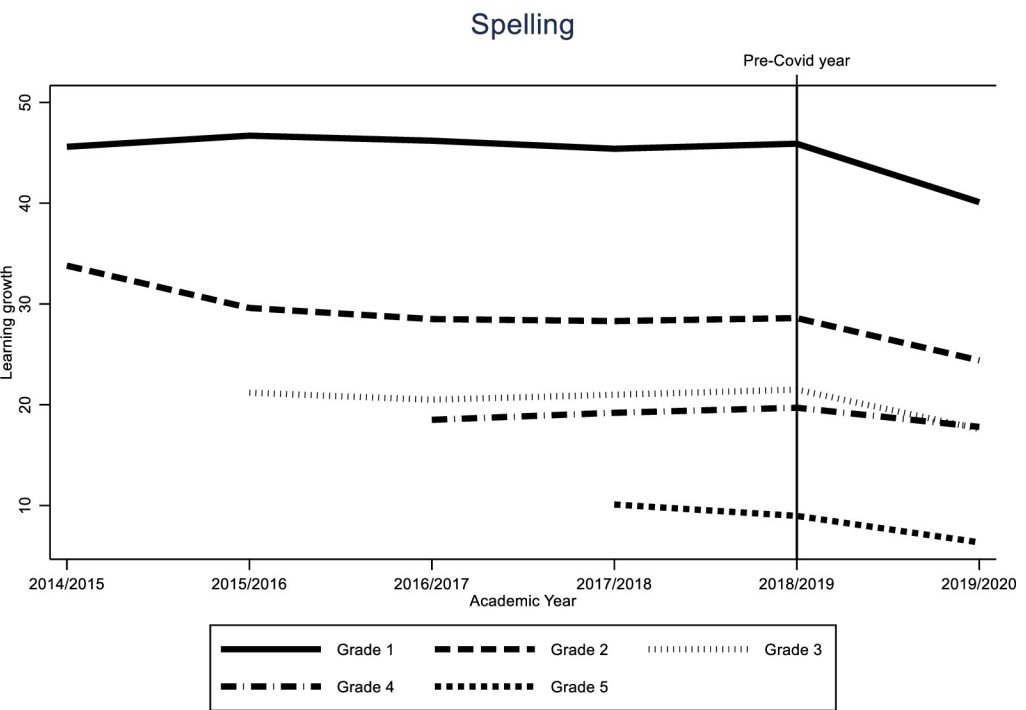

**Fig 6. Trends in learning gain over time–spelling.**

stable, we can be reasonably sure that the difference between the 2019/2020 cohort and the previous cohorts was caused by the impact of the pandemic. Figs 5–7 show the trends in learning gains per grade over time by plotting the (inverse probability population weighted) unstandardized learning gains for all available cohorts in reading, spelling, and math respectively. Because our data does not go back equally far for all grades, the lines are of different length. As noted earlier, we do not have information on grade 1 learning gains for the reading domain, as grade 1 students do not take a midterm test for this domain. For higher grades, we also have fewer available cohorts due to the manner in which data was collected (see also Table 1). Hence, for grade 2 we have data going back to academic year 2014/2015, for grade 3 from 2015/2016 onwards, for grade 4 starting in 2016/2017, and grade 5 starting 2017/2018.

The figures clearly show a marked decrease in learning gains between the COVID-19 affected cohort of the school year 2019/2020 relative to the prior cohorts in all domains and for most grades. For the spelling and math domains, learning gains prior to the COVID-19 cohorts were remarkably stable over time across grades 1 through 4. For grade 5, the 2018/2019 cohort had somewhat lower learning gains than the 2017/2018 cohort. However, since these are the only 2 pre-COVID-19 cohorts for which grade 5 data is available, it is unclear whether this represents a somewhat random fluctuation between cohorts, or whether it is part of a longer trend in declining grade 5 learning gain. For reading, the results are less clear. Grades 2 and 3 show a less stable trend over time than the other grades. For our main estimation sample of the 2017 and 2018 cohorts comprising the control group however, the differences in learning gain between these two cohorts is relatively small for all grades.

A different way of showing whether the 2019/2020 COVID-19 affected cohort is somewhat of an outlier in terms of their regular learning gain, is to plot the prior performance of this

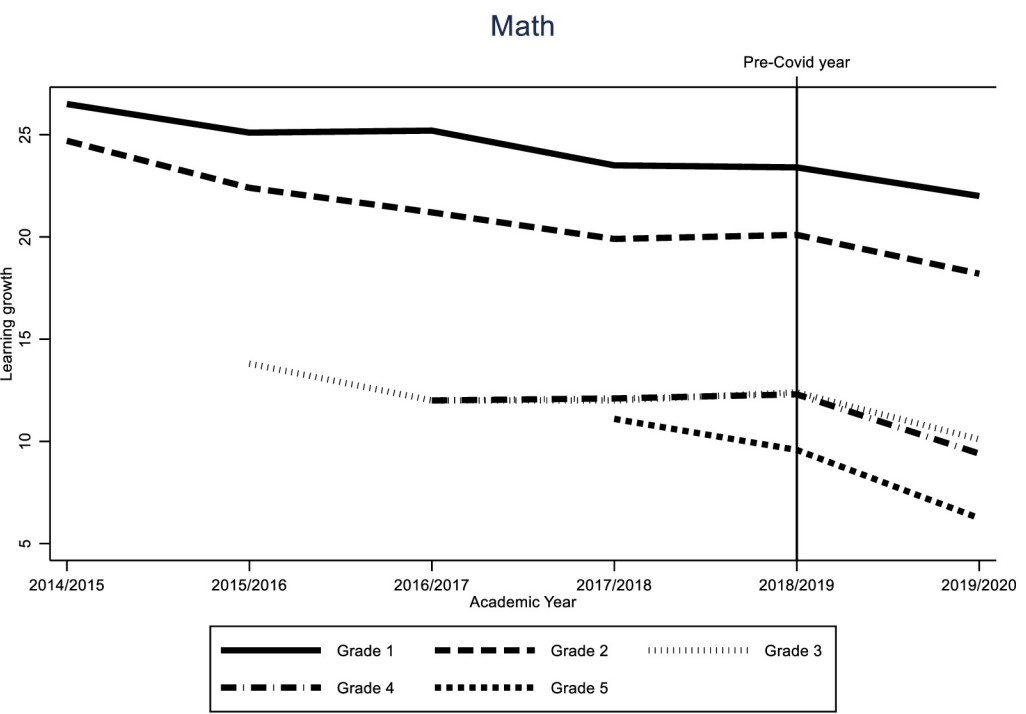

**Fig 7. Trends in learning gain over time–math.**

cohort during the years in which there was no pandemic and compare it to the performance of an earlier cohort. Figs 8–10 show the learning gain in all prior grades of the students who were in grade 5 during the 2019/2020 school year and those who were in grade 5 during the 2018/2019 school year for reading, spelling and math respectively. If the 2019/2020 cohort is relatively similar to the prior cohort, we should expect to see similar levels (and trends) of learning gain for grades 1 through 4. The pandemic occurred during grade 5 for the 2019/2020 cohort, and we should therefore expect lower levels of learning gain in grade 5 for the 2019/2020 cohort compared to the students that were in grade 5 during the 2018/2019 school year.

Looking at the figures, this is indeed what we see for spelling and math. Both cohorts are on remarkably similar learning gain trajectories from grade 1 through grade 4. In grade 5, the 2019/2020 cohort experiences a stronger decline in learning gain, especially in math, than the students of the previous cohort that were unaffected by the pandemic. For reading, the results are again less stable. The 2018/2019 cohort experienced a stronger decline in learning gain in grade 3 relative to the other grades and the 2019/2020 cohort. The overall pattern of decreasing learning gain from grade 2 to grade 3 and increasing learning gain from grade 3 to grade 4 is visible for both cohorts, however, and the grade 5 learning gain of the pandemic-affected 2019/2020 cohort does decrease more strongly than the learning gain of the prior, unaffected cohort.

## Conclusions

This study describes the additional inequality in learning gains of primary school students in the Netherlands during 12 weeks of disrupted learning due to the COVID-19-pandemic for three domains: reading, spelling and math. We show large inequalities in the learning loss

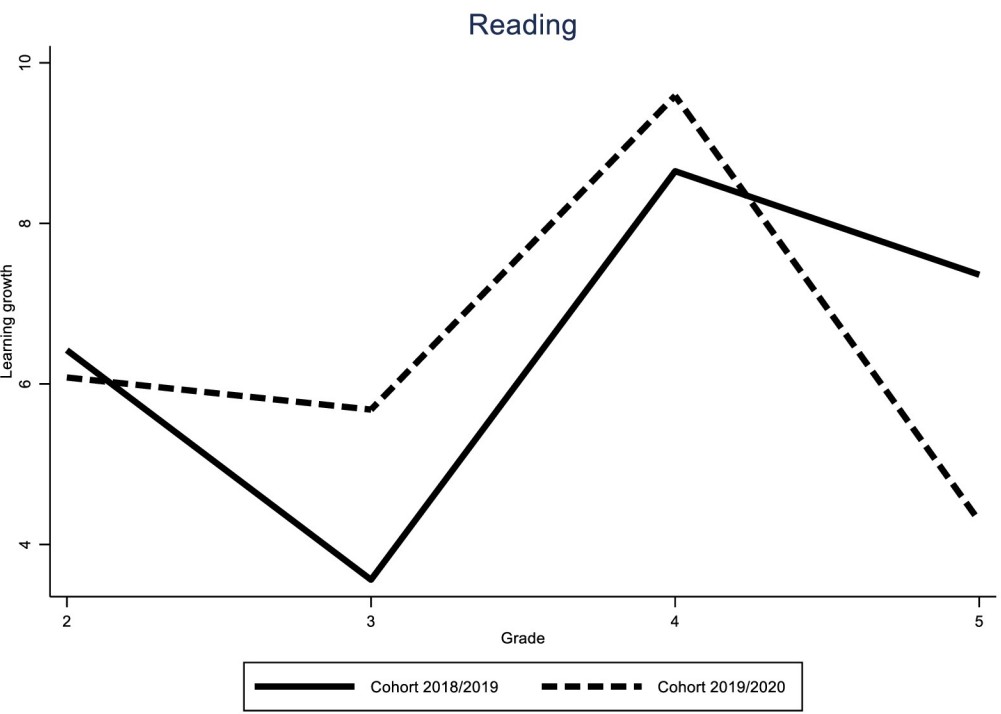

**Fig 8. Learning gain: Grade 5 students of 2018/2019 compared to Grade 5 students of 2019/2020 –reading.**

based on parental education and parental income, on top of already existing inequalities. The additional inequality in learning gains is largest among grades 2 and 3.

These results are quite alarming and indicate an *average delay* in learning of about 5.5 weeks for reading, and around 3 weeks for spelling and math with larger deficits in the higher grades. Relative to the period between the midterm- and end-of-term tests of around 20 weeks, this is rather a lot. It is to some extent reassuring that in general the decline in learning gains do not take place in the lowest grades in which the foundation for math and language skills are laid [34]. On the other hand, the decline in learning gains is larger for students from a low socioeconomic background (lower parental education and household income) for spelling an math and these inequalities are higher in the early grades. We see a delay of around 4 weeks for spelling and math for students with low-educated parents. This implies that during the school closure period students with low-educated parents hardly learned anything. However, there are no statistically significant socioeconomic status differences in reading scores. Previous research has shown that the home environment is important for the development of literacy skills and reading motivation [35, 36]. In line with this finding some have also suggested that center-based reading interventions might be less effective than mathematics interventions [37, 38]. Our finding that reading skills are less affected by the school closure support the idea that the family environment plays an important role in the development of reading skills, also when schools are open. In contrast, the limited increase in socioeconomic inequalities in reading skills is not in line with our expectations. Normally, family environment is an important source of inequality in reading skills [36] and we would expect inequalities to rise when schools closed and the role of family environment increased. We attribute the additional inequality in learning loss of students in math and spelling based on parental education and

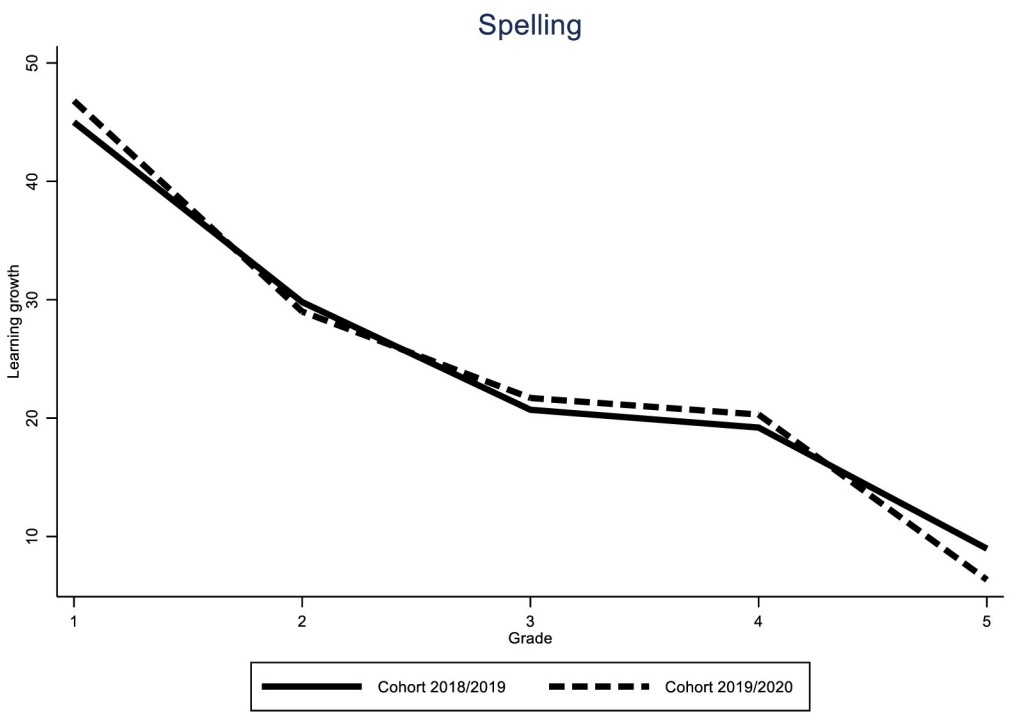

**Fig 9. Learning gain: Grade 5 students of 2018/2019 compared to Grade 5 students of 2019/2020 –spelling.**

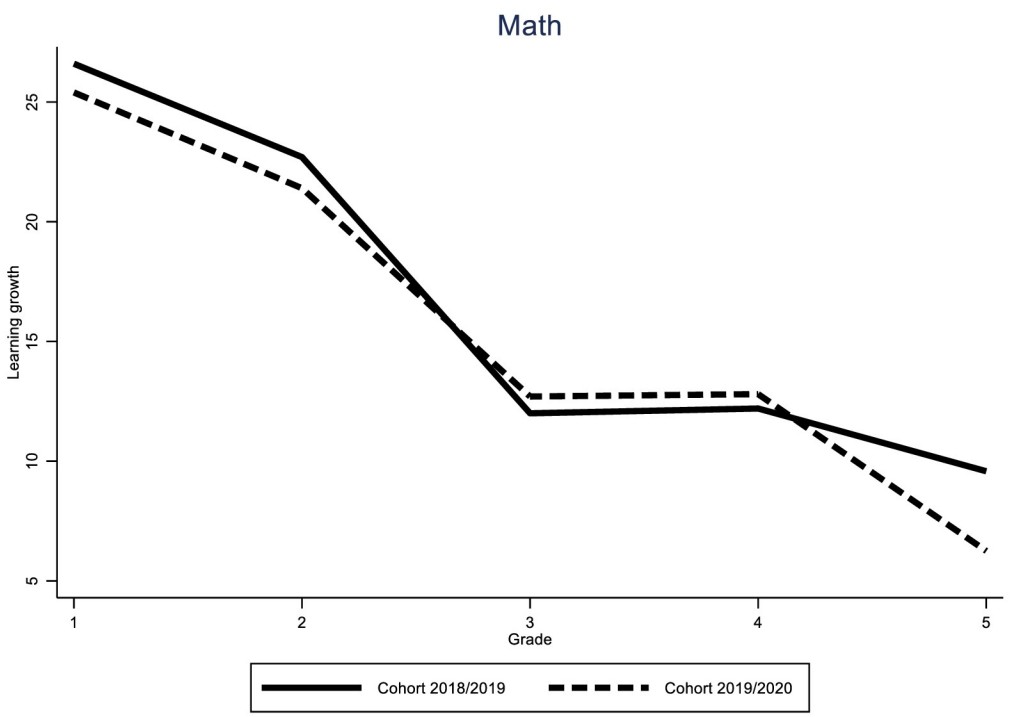

**Fig 10. Learning gain: Grade 5 students of 2018/2019 compared to Grade 5 students of 2019/2020 –math.**

**Table 5. The disparate impact of COVID-19 on student learning gains across students' household income and parental education levels–entropy weights.**

| VARIABLES | Reading | | | | Spelling | | | | | Math | | | | |
|---|---|---|---|---|---|---|---|---|---|---|---|---|---|---|
| | Grade 2 | Grade 3 | Grade 4 | Grade 5 | Grade 1 | Grade 2 | Grade 3 | Grade 4 | Grade 5 | Grade 1 | Grade 2 | Grade 3 | Grade 4 | Grade 5 |
| COVID-19 year (2019/2020) | -0.0898*** | -0.179*** | -0.139*** | -0.202*** | -0.265*** | -0.220*** | -0.193*** | -0.145*** | -0.176*** | -0.175*** | -0.289*** | -0.313*** | -0.382*** | -0.386*** |
| Household income | | | | | | | | | | | | | | |
| Medium | -0.0280** | -0.014 | -0.010 | 0.018 | -0.006 | -0.014 | -0.022 | -0.018 | -0.003 | -0.0265** | -0.0277** | -0.004 | -0.016 | -0.017 |
| High | 0.013 | 0.0310** | 0.0288* | 0.0358* | 0.002 | 0.019 | -0.0318* | -0.025 | 0.019 | -0.022 | -0.0476*** | 0.003 | 0.013 | -0.002 |
| Missing | 0.020 | -0.052 | 0.171 | -0.022 | -0.081 | 0.034 | 0.111 | 0.228* | 0.042 | 0.056 | -0.181 | -0.084 | -0.083 | 0.068 |
| COVID-19 year * Medium income household | 0.037 | 0.0641*** | 0.0423* | 0.039 | 0.019 | 0.0485* | 0.0421* | 0.005 | 0.023 | 0.0528** | 0.0935*** | 0.0699*** | 0.0567*** | 0.033 |
| COVID-19 year * High income household | 0.007 | 0.0502* | 0.032 | 0.037 | 0.033 | 0.0919*** | 0.0719** | 0.0669* | 0.004 | 0.0664** | 0.124*** | 0.0910*** | 0.044 | 0.016 |
| COVID-19 year * Household income missing | -0.087 | 0.142 | -0.213 | 0.258 | 0.293 | -0.057 | -0.283 | -0.072 | -0.182 | -0.262 | 0.300 | 0.199 | -0.043 | -0.298 |
| Parental education | | | | | | | | | | | | | | |
| Medium | 0.004 | 0.0456*** | 0.0358** | 0.028 | -0.007 | -0.019 | 0.018 | -0.019 | 0.0690*** | 0.0334** | -0.014 | -0.0314* | 0.001 | 0.004 |
| High | 0.022 | 0.118*** | 0.109*** | 0.0431** | -0.0409** | 0.027 | 0.0364** | -0.015 | 0.115*** | 0.005 | -0.007 | -0.005 | 0.006 | 0.016 |
| Missing | -0.0368* | 0.0485** | 0.029 | 0.0605*** | -0.016 | 0.009 | 0.0481** | -0.007 | 0.0596** | 0.027 | -0.022 | -0.030 | 0.019 | 0.020 |
| COVID-19 year * Parental education medium | -0.040 | -0.0551* | -0.0555* | -0.049 | 0.049 | 0.045 | -0.027 | 0.030 | -0.035 | -0.032 | 0.047 | 0.0657** | 0.034 | 0.023 |
| COVID-19 year * Parental education high | 0.009 | -0.028 | 0.001 | -0.013 | 0.119*** | 0.0698** | 0.039 | 0.054 | 0.011 | 0.016 | 0.119*** | 0.106*** | 0.132*** | 0.046 |
| COVID-19 year * Parental education missing | 0.050 | -0.044 | -0.003 | -0.0801* | 0.0921** | 0.011 | 0.006 | 0.017 | 0.027 | 0.003 | 0.130*** | 0.134*** | 0.038 | 0.042 |
| Additional controls | Yes | Yes | Yes | Yes | Yes | Yes | Yes | Yes | Yes | Yes | Yes | Yes | Yes | Yes |
| Constant | 0.0523*** | 0.0202 | -0.0470** | 0.0290 | 0.0895*** | 0.0142 | 0.00339 | 0.0340* | -0.0418 | 0.0521*** | 0.0617*** | 0.0347* | 0.0385*** | 0.0548*** |
| Observations | 54,937 | 68,272 | 68,347 | 49,158 | 62,643 | 66,810 | 71,630 | 70,610 | 47,764 | 70,385 | 70,668 | 75,043 | 79,239 | 69,534 |
| R-squared | 0.002 | 0.013 | 0.007 | 0.012 | 0.012 | 0.012 | 0.008 | 0.003 | 0.012 | 0.006 | 0.005 | 0.008 | 0.017 | 0.030 |
| Clusters | 1038 | 1154 | 1149 | 1088 | 1148 | 1161 | 1158 | 1153 | 1076 | 1159 | 1167 | 1166 | 1171 | 1159 |

The outcome variable, learning gain between the midterm and the end-of-term test, has been standardized within grade using the inverse probability population weighted means and standard deviations. The baseline category for "COVID-19 year" are the pooled students from the 2017/2018 and 2018/2019 school years. The baseline category for household income is "low". The baseline category for parental education is "low". Additional controls include student gender, students' migration background, and a dummy indicating whether students took their end-of-term test at the start of the next, rather than at the end of the current school year. Observations are weighted using entropy weights. Standard errors are clustered at the school level and are omitted for brevity.

* p<0.10; ** p<0.05; *** p<0.01.

**Table 6. The disparate impact of COVID-19 on student learning gains across students' household income and parental education levels–unweighted.**

| VARIABLES | Reading | | | | Spelling | | | | | Math | | | | |
|---|---|---|---|---|---|---|---|---|---|---|---|---|---|---|
| | Grade 2 | Grade 3 | Grade 4 | Grade 5 | Grade 1 | Grade 2 | Grade 3 | Grade 4 | Grade 5 | Grade 1 | Grade 2 | Grade 3 | Grade 4 | Grade 5 |
| COVID-19 year (2019/2020) | -0.0931*** | -0.180*** | -0.137*** | -0.205*** | -0.266*** | -0.211*** | -0.194*** | -0.146*** | -0.181*** | -0.179*** | -0.285*** | -0.313*** | -0.383*** | -0.378*** |
| Household income | | | | | | | | | | | | | | |
| Medium | -0.0233* | -0.0105 | -0.00709 | 0.0186 | -0.00348 | -0.0156 | -0.0212 | -0.0161 | 0.00460 | -0.0262** | -0.0247** | -0.00393 | -0.0162 | -0.0144 |
| High | 0.0181 | 0.0326** | 0.0291* | 0.0327* | 0.00653 | 0.0176 | -0.0323** | -0.0214 | 0.0145 | -0.0214 | -0.0427** | 0.00333 | 0.0120 | -0.00313 |
| Missing | 0.0498 | 0.0456 | 0.111* | 0.117 | 0.0560 | 0.207*** | 0.0962 | 0.284*** | 0.213** | -0.0534 | 0.0372 | 0.0135 | 0.0834 | 0.0846 |
| COVID-19 year * Medium income household | 0.0352 | 0.0632*** | 0.0403* | 0.0397 | 0.0185 | 0.0478** | 0.0419* | 0.00458 | 0.0168 | 0.0541** | 0.0934*** | 0.0727*** | 0.0570*** | 0.0312 |
| COVID-19 year * High income household | 0.00549 | 0.0516* | 0.0330 | 0.0430 | 0.0306 | 0.0911*** | 0.0724** | 0.0647** | 0.00992 | 0.0681** | 0.122*** | 0.0940*** | 0.0441 | 0.0186 |
| COVID-19 year * Household income missing | -0.0243 | 0.0611 | -0.0627 | 0.197* | 0.00978 | 0.0350 | 0.159 | -0.0180 | -0.190 | 0.167* | -0.0218 | 0.0881 | 0.0165 | -0.0434 |
| Parental education | | | | | | | | | | | | | | |
| Medium | 0.00413 | 0.0461*** | 0.0362** | 0.0209 | -0.00865 | -0.0180 | 0.0156 | -0.0160 | 0.0832*** | 0.0293* | -0.0106 | -0.0249 | 0.00206 | 0.00977 |
| High | 0.0235 | 0.118*** | 0.110*** | 0.0467** | -0.0426** | 0.0315* | 0.0329* | -0.0142 | 0.127*** | 0.00240 | -0.00397 | -0.00166 | 0.00610 | 0.0153 |
| Missing | -0.0303 | 0.0483** | 0.0301 | 0.0459** | -0.0150 | 0.0127 | 0.0450** | -0.0115 | 0.0680*** | 0.0233 | -0.0180 | -0.0241 | 0.0169 | 0.0181 |
| COVID-19 year * Parental education medium | -0.0321 | -0.0530* | -0.0560* | -0.0416 | 0.0524 | 0.0440 | -0.0248 | 0.0286 | -0.0484 | -0.0279 | 0.0428 | 0.0626** | 0.0339 | 0.0162 |
| COVID-19 year * Parental education high | 0.0148 | -0.0253 | 0.00250 | -0.0153 | 0.122*** | 0.0660* | 0.0421 | 0.0548 | -0.000224 | 0.0192 | 0.117*** | 0.106*** | 0.133*** | 0.0470 |
| COVID-19 year * Parental education missing | 0.0536 | -0.0418 | -0.00150 | -0.0660 | 0.0919** | 0.00363 | 0.0141 | 0.0256 | 0.0190 | 0.00916 | 0.118*** | 0.133*** | 0.0498 | 0.0501 |
| Additional controls | Yes | Yes | Yes | Yes | Yes | Yes | Yes | Yes | Yes | Yes | Yes | Yes | Yes | Yes |
| Constant | 0.0398** | 0.0108 | -0.0539*** | 0.0244 | 0.0866*** | 0.00969 | 0.00400 | 0.0325* | -0.0412* | 0.0488*** | 0.0524*** | 0.0256 | 0.0365** | 0.0430** |
| Observations | 55,303 | 68,721 | 68,802 | 49,491 | 63,148 | 67,295 | 72,125 | 71,089 | 48,093 | 70,969 | 71,188 | 75,577 | 79,786 | 69,986 |
| R-squared | 0.002 | 0.011 | 0.005 | 0.010 | 0.010 | 0.010 | 0.007 | 0.004 | 0.012 | 0.005 | 0.004 | 0.007 | 0.014 | 0.022 |
| Clusters | 1038 | 1154 | 1149 | 1089 | 1148 | 1161 | 1158 | 1153 | 1079 | 1159 | 1167 | 1166 | 1171 | 1159 |

The outcome variable, learning gain between the midterm and the end-of-term test, has been standardized within grade using the inverse probability population weighted means and standard deviations. The baseline category for "COVID-19 year" are the pooled students from the 2017/2018 and 2018/2019 school years. The baseline category for household income is "low". The baseline category for parental education is "low". Additional controls include student gender, students' migration background, and a dummy indicating whether students took their end-of-term test at the start of the next, rather than at the end of the current school year. Standard errors are clustered at the school level and are omitted for brevity.

* p<0.10; ** p<0.05; *** p<0.01.

**Table 7. The disparate impact of COVID-19 on student learning gains across students' household income and parental education levels–no additional controls.**

| VARIABLES | Reading | | | | Spelling | | | | | Math | | | | |
|---|---|---|---|---|---|---|---|---|---|---|---|---|---|---|
| | Grade 2 | Grade 3 | Grade 4 | Grade 5 | Grade 1 | Grade 2 | Grade 3 | Grade 4 | Grade 5 | Grade 1 | Grade 2 | Grade 3 | Grade 4 | Grade 5 |
| COVID-19 year (2019/2020) | -0.0574 | -0.135*** | -0.0983*** | -0.214*** | -0.319*** | -0.259*** | -0.211*** | -0.153*** | -0.210*** | -0.167*** | -0.304*** | -0.273*** | -0.338*** | -0.386*** |
| Household income | | | | | | | | | | | | | | |
| Medium | -0.0211 | 0.00699 | -0.00942 | 0.00298 | -0.00724 | -0.0488*** | -0.0305* | -0.0300** | 0.00617 | -0.0394*** | -0.0381*** | -0.0247** | -0.0258** | -0.0282** |
| High | 0.0209 | 0.0552*** | 0.0273* | 0.00770 | 0.00661 | -0.0165 | -0.0461*** | -0.0328* | 0.0175 | -0.0384*** | -0.0579*** | -0.0174 | -0.00379 | -0.0196 |
| Missing | 0.0503 | -0.0110 | 0.150 | -0.0201 | -0.0361 | 0.0921 | 0.167 | 0.237* | 0.0643 | 0.0663 | -0.102 | -0.157 | -0.152 | 0.0412 |
| COVID-19 year * Medium income household | 0.0365 | 0.0638*** | 0.0308 | 0.0599* | 0.0230 | 0.0583*** | 0.0402 | -0.00621 | 0.0134 | 0.0395 | 0.0974*** | 0.0738*** | 0.0515*** | 0.0411 |
| COVID-19 year * High income household | -0.0106 | 0.0421 | 0.0171 | 0.0857** | 0.0313 | 0.114*** | 0.0753** | 0.0507* | -0.00855 | 0.0616** | 0.128*** | 0.0910*** | 0.0337 | 0.0194 |
| COVID-19 year * Household income missing | 0.0252 | 0.135 | -0.205 | 0.277 | 0.300 | -0.0460 | -0.311 | -0.0846 | -0.145 | -0.227 | 0.238 | 0.210 | 0.0673 | -0.278 |
| Parental education | | | | | | | | | | | | | | |
| Medium | 0.00397 | 0.0672*** | 0.0281* | 0.00920 | -0.0275 | -0.0510*** | -0.00370 | -0.0382** | 0.0926*** | 0.00955 | -0.0261 | -0.0394** | -0.0103 | 0.00714 |
| High | 0.0287 | 0.143*** | 0.0974*** | 0.0403* | -0.0569*** | -0.00829 | 0.0112 | -0.0365* | 0.131*** | -0.0187 | -0.0206 | -0.0195 | -0.0117 | 0.00263 |
| Missing | -0.0305 | 0.0684*** | 0.0255 | 0.0426* | -0.0184 | -0.0193 | 0.0307 | -0.0362 | 0.0741*** | 0.00600 | -0.0336 | -0.0377* | -0.00488 | 0.00760 |
| COVID-19 year * Parental education medium | -0.0477 | -0.0599* | -0.0496 | -0.0473 | 0.0640* | 0.0310 | -0.0244 | 0.0446 | -0.0363 | -0.0177 | 0.0649* | 0.0434 | 0.0237 | 0.0198 |
| COVID-19 year * Parental education high | 0.00326 | -0.0405 | 0.0109 | -0.0379 | 0.130*** | 0.0599* | 0.0431 | 0.0651* | 0.0502 | 0.0173 | 0.128*** | 0.0841*** | 0.120*** | 0.0508 |
| COVID-19 year * Parental education missing | 0.0323 | -0.0495 | -0.000200 | -0.0896* | 0.105** | 0.0121 | 0.0142 | 0.0386 | 0.0520 | 0.00873 | 0.137*** | 0.108*** | 0.0368 | 0.0253 |
| Additional controls | No | No | No | No | No | No | No | No | No | No | No | No | No | No |
| Constant | 0.0111 | -0.0707*** | -0.0313* | 0.0372* | 0.108*** | 0.105*** | 0.0709*** | 0.0884*** | -0.0445* | 0.0775*** | 0.0968*** | 0.0912*** | 0.100*** | 0.125*** |
| Observations | 54,937 | 68,272 | 68,347 | 49,159 | 62,643 | 66,810 | 71,630 | 70,610 | 47,765 | 70,385 | 70,668 | 75,043 | 79,239 | 69,535 |
| R-squared | 0.001 | 0.008 | 0.004 | 0.010 | 0.010 | 0.007 | 0.006 | 0.003 | 0.010 | 0.004 | 0.005 | 0.006 | 0.013 | 0.024 |
| Clusters | 1038 | 1154 | 1149 | 1088 | 1148 | 1161 | 1158 | 1153 | 1076 | 1159 | 1167 | 1166 | 1171 | 1159 |

The outcome variable, learning gain between the midterm and the end-of-term test, has been standardized within grade using the inverse probability population weighted means and standard deviations. The baseline category for "COVID-19 year" are the pooled students from the 2017/2018 and 2018/2019 school years. The baseline category for household income is "low". The baseline category for parental education is "low". Observations are weighted using inverse probability weights. Standard errors are clustered at the school level and are omitted for brevity.

* p<0.10; ** p<0.05; *** p<0.01.

**Table 8. The disparate impact of COVID-19 on student learning gains across students' household income and parental education levels–additional control for prior performance.**

| VARIABLES | Reading | | | Spelling | | | | Math | | | |
|---|---|---|---|---|---|---|---|---|---|---|---|
| | Grade 3 | Grade 4 | Grade 5 | Grade 2 | Grade 3 | Grade 4 | Grade 5 | Grade 2 | Grade 3 | Grade 4 | Grade 5 |
| COVID-19 year (2019/2020) | -0.167*** | -0.136*** | -0.225*** | -0.225*** | -0.226*** | -0.153*** | -0.164*** | -0.302*** | -0.298*** | -0.358*** | -0.401*** |
| Household income | | | | | | | | | | | |
| Medium | -0.00851 | -0.00532 | 0.00698 | -0.00249 | -0.0149 | -0.00630 | 0.0185 | -0.0219* | -0.00227 | -0.00700 | -0.0152 |
| High | 0.0410** | 0.0287* | 0.0173 | 0.0292 | -0.0232 | -0.00466 | 0.0329 | -0.0383*** | 0.00723 | 0.0136 | -0.00481 |
| Missing | -0.0720 | 0.186* | -0.0851 | 0.0679 | 0.121 | 0.256** | 0.140 | -0.0993 | -0.240** | -0.233* | 0.111 |
| COVID-19 year * Medium income household | 0.0726*** | 0.0322 | 0.0720** | 0.0311 | 0.0567** | -0.0102 | -0.00369 | 0.0952*** | 0.0759*** | 0.0374 | 0.0523* |
| COVID-19 year * High income household | 0.0258 | 0.0275 | 0.0951** | 0.0904*** | 0.0873*** | 0.0455 | -0.0447 | 0.134*** | 0.101*** | 0.0313 | 0.0372 |
| COVID-19 year * Household income missing | 0.150 | -0.0723 | 0.319 | -0.108 | -0.347 | -0.116 | -0.161 | 0.0673 | 0.339 | 0.242 | -0.307 |
| Parental education | | | | | | | | | | | |
| Medium | 0.0372** | 0.0356* | 0.00441 | -0.0251 | 0.0109 | -0.0252 | 0.104*** | -0.0108 | -0.0303* | 0.00739 | 0.0121 |
| High | 0.107*** | 0.0976*** | 0.0267 | 0.0204 | 0.0322* | -0.0173 | 0.146*** | -0.00229 | -0.00404 | 0.00797 | 0.0155 |
| Missing | 0.0525** | 0.0325 | 0.0177 | -0.0149 | 0.0393* | -0.0388 | 0.0778*** | -0.0138 | -0.0339 | 0.00453 | 0.0182 |
| COVID-19 year * Parental education medium | -0.0633* | -0.0498 | -0.0296 | 0.0414 | -0.0255 | 0.0416 | -0.0341 | 0.0539 | 0.0438 | 0.0161 | 0.0257 |
| COVID-19 year * Parental education high | -0.0214 | 0.0181 | -0.0264 | 0.0720* | 0.0403 | 0.0711** | 0.0425 | 0.118*** | 0.0861** | 0.118*** | 0.0479 |
| COVID-19 year * Parental education missing | -0.0914** | 0.00214 | -0.0711 | 0.0161 | 0.000974 | 0.0456 | 0.0415 | 0.116*** | 0.101** | 0.0308 | 0.0184 |
| Learning gain in the prior year | 0.0610*** | 0.0474*** | 0.0101 | 0.0926*** | 0.0584*** | 0.0589*** | 0.0124 | 0.105*** | 0.0368*** | 0.0386*** | 0.00947 |
| Additional controls | Yes | Yes | Yes | Yes | Yes | Yes | Yes | Yes | Yes | Yes | Yes |
| Constant | -0.0326 | -0.0807*** | 0.0470* | -0.0549** | -0.0355* | -0.00157 | -0.0891*** | -0.00413 | 0.00839 | 0.0113 | 0.0406** |
| Observations | 53,421 | 59,728 | 41,762 | 56,806 | 63,006 | 63,631 | 42,531 | 65,911 | 69,730 | 73,756 | 64,950 |
| R-squared | 0.012 | 0.006 | 0.011 | 0.014 | 0.010 | 0.004 | 0.012 | 0.008 | 0.007 | 0.015 | 0.028 |
| Clusters | 1072 | 1112 | 1020 | 1151 | 1147 | 1135 | 1033 | 1161 | 1159 | 1166 | 1151 |

The outcome variable, learning gain between the midterm and the end-of-term test, has been standardized within grade using the inverse probability population weighted means and standard deviations. The baseline category for "COVID-19 year" are the pooled students from the 2017/2018 and 2018/2019 school years. The baseline category for household income is "low". The baseline category for parental education is "low". Additional controls include student gender, students' migration background, and a dummy indicating whether students took their end-of-term test at the start of the next, rather than at the end of the current school year. Observations are weighted using inverse probability weights. Standard errors are clustered at the school level and are omitted for brevity.

* p<0.10; ** p<0.05; *** p<0.01.

**Table 9. The disparate impact of COVID-19 on student learning gains across students' household income and parental education levels–school fixed effects.**

| VARIABLES | Reading | | | | Spelling | | | | | Math | | | | |
|---|---|---|---|---|---|---|---|---|---|---|---|---|---|---|
| | Grade 2 | Grade 3 | Grade 4 | Grade 5 | Grade 1 | Grade 2 | Grade 3 | Grade 4 | Grade 5 | Grade 1 | Grade 2 | Grade 3 | Grade 4 | Grade 5 |
| COVID-19 year (2019/2020) | -0.0798** | -0.166*** | -0.145*** | -0.230*** | -0.283*** | -0.249*** | -0.217*** | -0.156*** | -0.239*** | -0.195*** | -0.313*** | -0.305*** | -0.367*** | -0.406*** |
| Household income | | | | | | | | | | | | | | |
| Medium | -0.0245* | -0.0172 | -0.00646 | 0.0200 | -0.00534 | -0.0160 | -0.0121 | 0.00489 | 0.00645 | -0.0170 | -0.0246** | -0.00461 | -0.0113 | -0.00903 |
| High | 0.0138 | 0.0194 | 0.0212 | 0.0295 | 0.00519 | -0.00835 | -0.0306* | -0.0108 | 0.0199 | -0.0282* | -0.0361* | 0.000759 | 0.00281 | -0.00830 |
| Missing | 0.0428 | 0.0197 | 0.132 | -0.0845 | -0.0153 | 0.0604 | 0.135 | 0.218* | 0.0805 | 0.0883 | -0.0231 | -0.202* | -0.123 | -0.0421 |
| COVID-19 year * Medium income household | 0.0312 | 0.0569** | 0.0296 | 0.0463 | 0.0179 | 0.0684*** | 0.0452* | -0.0153 | 0.0274 | 0.0351 | 0.0973*** | 0.0670*** | 0.0519** | 0.0445* |
| COVID-19 year * High income household | -0.00824 | 0.0410 | 0.0129 | 0.0802** | 0.0129 | 0.119*** | 0.0738** | 0.0343 | -0.0111 | 0.0598** | 0.130*** | 0.0921*** | 0.0445 | 0.0173 |
| COVID-19 year * Household income missing | 0.0564 | 0.115 | -0.178 | 0.309 | 0.238 | -0.0885 | -0.260 | -0.170 | -0.225 | -0.178 | 0.256 | 0.354* | 0.0647 | -0.247 |
| Parental education | | | | | | | | | | | | | | |
| Medium | -0.00356 | 0.0374** | 0.0298* | 0.0238 | -0.0287 | -0.0135 | 0.00889 | -0.0256 | 0.0606*** | 0.0267 | -0.00748 | -0.0164 | 0.00792 | 0.0195 |
| High | 0.0116 | 0.0847*** | 0.0817*** | 0.0531** | -0.0814*** | 0.0133 | 0.0332* | -0.0408** | 0.0888*** | -0.00754 | 0.000545 | -0.00401 | -0.000905 | 0.00222 |
| Missing | -0.0350 | 0.0316 | 0.0298 | 0.0585** | -0.0271 | 0.0251 | 0.0497** | -0.00579 | 0.0378 | 0.0152 | -0.0187 | -0.0116 | 0.0134 | 0.0198 |
| COVID-19 year * Parental education medium | -0.0455 | -0.0716** | -0.0514 | -0.0392 | 0.0624* | 0.0264 | -0.0182 | 0.0408 | -0.0379 | -0.00794 | 0.0603* | 0.0414 | 0.00609 | 0.0219 |
| COVID-19 year * Parental education high | 0.00297 | -0.0584* | 0.00742 | -0.0263 | 0.139*** | 0.0459 | 0.0390 | 0.0618* | 0.0315 | 0.0212 | 0.128*** | 0.0786*** | 0.103*** | 0.0514 |
| COVID-19 year * Parental education missing | 0.0252 | -0.0515 | -0.000714 | -0.0824* | 0.103** | 0.000242 | 0.00561 | 0.0325 | 0.0483 | 0.00728 | 0.131*** | 0.101** | 0.0160 | 0.0306 |
| Additional controls | Yes | Yes | Yes | Yes | Yes | Yes | Yes | Yes | Yes | Yes | Yes | Yes | Yes | Yes |
| School Fixed Effects | Yes | Yes | Yes | Yes | Yes | Yes | Yes | Yes | Yes | Yes | Yes | Yes | Yes | Yes |
| Constant | 0.0517*** | 0.0392** | -0.0374** | 0.0310 | 0.113*** | 0.0252 | 0.00265 | 0.0383** | -0.00487 | 0.0621*** | 0.0489*** | 0.0285* | 0.0422** | 0.0531*** |
| Observations | 54,937 | 68,272 | 68,347 | 49,158 | 62,643 | 66,810 | 71,630 | 70,610 | 47,764 | 70,385 | 70,668 | 75,043 | 79,239 | 69,534 |
| R-squared | 0.038 | 0.052 | 0.044 | 0.061 | 0.074 | 0.106 | 0.065 | 0.101 | 0.106 | 0.060 | 0.060 | 0.055 | 0.061 | 0.099 |
| Clusters | 1038 | 1154 | 1149 | 1088 | 1148 | 1161 | 1158 | 1153 | 1076 | 1159 | 1167 | 1166 | 1171 | 1159 |

The outcome variable, learning gain between the midterm and the end-of-term test, has been standardized within grade using the inverse probability population weighted means and standard deviations. The baseline category for "COVID-19 year" are the pooled students from the 2017/2018 and 2018/2019 school years. The baseline category for household income is "low". The baseline category for parental education is "low". Additional controls include student gender, students' migration background, and a dummy indicating whether students took their end-of-term test at the start of the next, rather than at the end of the current school year. Observations are weighted using inverse probability weights. Standard errors are clustered at the school level and are omitted for brevity.

* p<0.10; ** p<0.05; *** p<0.01.

**Table 10. The disparate impact of COVID-19 on student learning gains across students' household income and parental education levels–pooled over all grades.**

| VARIABLES | Entropy weights | | | Unweighted | | | No additional controls | | | Control for prior performance | | | School Fixed Effects | | |
|---|---|---|---|---|---|---|---|---|---|---|---|---|---|---|---|
| | Reading | Spelling | Math | Reading | Spelling | Math | Reading | Spelling | Math | Reading | Spelling | Math | Reading | Spelling | Math |
| COVID-19 year (2019/2020) | -0.158*** | -0.202*** | -0.315*** | -0.158*** | -0.202*** | -0.315*** | -0.122*** | -0.228*** | -0.294*** | -0.173*** | -0.199*** | -0.343*** | -0.153*** | -0.223*** | -0.324*** |
| Household income | | | | | | | | | | | | | | | |
| Medium | -0.00707 | -0.0126* | -0.0180*** | -0.00597 | -0.0126* | -0.0170*** | -0.00454 | -0.0258*** | -0.0304*** | -0.000723 | -0.00479 | -0.0111* | -0.00777 | -0.00637 | 0.0139*** |
| High | 0.0291*** | -0.00500 | -0.00862 | 0.0290*** | -0.00577 | -0.00790 | 0.0325*** | -0.0189** | -0.0238*** | 0.0333*** | 0.00392 | -0.00315 | 0.0216** | -0.00897 | -0.0119* |
| Missing | 0.0568 | 0.0784 | -0.0423 | 0.0773** | 0.168*** | 0.0292 | 0.0535 | 0.126** | -0.0622 | 0.0266 | 0.153** | -0.111* | 0.0452 | 0.124** | -0.0501 |
| COVID-19 year * Medium income household | 0.0449*** | 0.0266** | 0.0605*** | 0.0443*** | 0.0270* | 0.0607*** | 0.0451*** | 0.0272** | 0.0587*** | 0.0520*** | 0.0201 | 0.0652*** | 0.0411*** | 0.0254*** | 0.0573*** |
| COVID-19 year * High income household | 0.0312** | 0.0575*** | 0.0637*** | 0.0320** | 0.0591*** | 0.0638*** | 0.0250 | 0.0610*** | 0.0603*** | 0.0402* | 0.0553*** | 0.0721*** | 0.0268* | 0.0540*** | 0.0611*** |
| COVID-19 year * Household income missing | -0.0106 | -0.0283 | -0.0276 | 0.0398 | 0.00728 | 0.0569 | 0.0352 | -0.0400 | 0.0140 | 0.124 | -0.139 | 0.0941 | 0.0371 | -0.0612 | 0.0404 |
| Parental education | | | | | | | | | | | | | | | |
| Medium | 0.0256*** | 0.00866 | -0.00143 | 0.0280*** | 0.00895 | -0.000243 | 0.0289*** | -0.00929 | -0.0141* | 0.0258** | 0.00859 | -0.00645 | 0.0233*** | -0.000732 | 0.00337 |
| High | 0.0747*** | 0.0217** | -0.000638 | 0.0784*** | 0.0231*** | 0.00186 | 0.0813*** | 0.00476 | -0.0158* | 0.0810*** | 0.0374*** | 0.00310 | 0.0601*** | 0.00105 | -0.00476 |
| Missing | 0.0251** | 0.0194* | 0.00701 | 0.0255** | 0.0185* | 0.00514 | 0.0305*** | 0.00337 | -0.0101 | 0.0343*** | 0.0115 | -0.00309 | 0.0231** | 0.0174* | 0.00382 |
| COVID-19 year * Parental education medium | -0.0446*** | 0.0111 | 0.0318** | -0.0453*** | 0.0119 | 0.0321* | -0.0484*** | 0.0166 | 0.0335* | -0.0474* | 0.0110 | 0.0352** | -0.0558*** | 0.0184 | 0.0315** |
| COVID-19 year * Parental education high | -0.00200 | 0.0592*** | 0.0942*** | -0.00349 | 0.0589*** | 0.0938*** | -0.01000 | 0.0635*** | 0.0893*** | -0.00436 | 0.0570*** | 0.0982*** | 0.0171 | 0.0629*** | 0.0883*** |
| COVID-19 year * Parental education missing | -0.0204 | 0.0306 | 0.0616*** | -0.0190 | 0.0337* | 0.0662*** | -0.0273 | 0.0393** | 0.0573*** | -0.0431* | 0.0277 | 0.0582*** | -0.0300 | 0.0340* | 0.0555*** |
| Learning gain in the prior year | . | . | . | . | . | . | . | . | . | 0.0412*** | 0.0582*** | 0.0465*** | | | |

(Continued)

**Table 10.** (Continued)

| VARIABLES | Entropy weights | | | Unweighted | | | No additional controls | | | Control for prior performance | | | School Fixed Effects | | |
|---|---|---|---|---|---|---|---|---|---|---|---|---|---|---|---|
| | Reading | Spelling | Math | Reading | Spelling | Math | Reading | Spelling | Math | Reading | Spelling | Math | Reading | Spelling | Math |
| Additional controls | Yes | Yes | Yes | Yes | Yes | Yes | No | No | No | Yes | Yes | Yes | Yes | Yes | Yes |
| School Fixed Effects | No | No | No | No | No | No | No | No | No | No | No | No | Yes | Yes | Yes |
| Constant | 0.0152 | 0.0144 | 0.0548*** | -0.00171 | 0.0193 | 0.0390*** | -0.0196** | 0.0698*** | 0.0983*** | -0.0406*** | -0.0423*** | 0.0113 | 0.0158 | 0.0319*** | 0.0460*** |
| Observations | 240,714 | 319,457 | 364,869 | 242,317 | 321,750 | 367,506 | 240,715 | 319,458 | 364,870 | 154,911 | 225,974 | 274,347 | 240,715 | 319,458 | 364,870 |
| R-squared | 0.005 | 0.007 | 0.009 | 0.005 | 0.006 | 0.009 | 0.005 | 0.006 | 0.008 | 0.007 | 0.007 | 0.012 | 0.020 | 0.034 | 0.026 |
| Clusters | 1174 | 1172 | 1178 | 1174 | 1172 | 1178 | 1174 | 1172 | 1178 | 1148 | 1163 | 1174 | 1174 | 1172 | 1178 |

Note: the outcome variable, learning gain between the midterm and the end-of-term test, has been standardized within grade using the inverse probability population weighted means and standard deviations. The baseline category for "COVID-19 year" are the pooled students from the 2017/2018 and 2018/2019 school years. The baseline category for household income is "low". The baseline category for parental education is "low". Additional controls include student gender, students' migration background, an indicator for student grade, and a dummy indicating whether students took their end-of-term test at the start of the next, rather than at the end of the current school year. Standard errors are clustered at the school level and are omitted for brevity.

* p<0.10; ** p<0.05; *** p<0.01.

**Table 11. Underlying regression results main effects (Fig 1).**

| VARIABLES | Reading | | | | Spelling | | | | | Math | | | | |
|---|---|---|---|---|---|---|---|---|---|---|---|---|---|---|
| | Grade 2 | Grade 3 | Grade 4 | Grade 5 | Grade 1 | Grade 2 | Grade 3 | Grade 4 | Grade 5 | Grade 1 | Grade 2 | Grade 3 | Grade 4 | Grade 5 |
| COVID-19 year (2019/2020) | -0.0632*** | -0.163*** | -0.116*** | -0.200*** | -0.166*** | -0.129*** | -0.152*** | -0.0977*** | -0.179*** | -0.131*** | -0.129*** | -0.166*** | -0.265*** | -0.326*** |
| Parental education | | | | | | | | | | | | | | |
| Medium | -0.0160 | 0.0240 | 0.0142 | 0.00377 | -0.00251 | -0.00756 | 0.00368 | -0.00735 | 0.0647*** | 0.0226 | 0.00628 | -0.00807 | 0.00767 | 0.0248 |
| High | 0.0255 | 0.102*** | 0.105*** | 0.0397** | -0.0107 | 0.0495*** | 0.0450*** | 0.00602 | 0.129*** | 0.00857 | 0.0354** | 0.0300** | 0.0427*** | 0.0350** |
| Missing | -0.0246 | 0.0309* | 0.0286* | 0.0241 | 0.0156 | 0.00977 | 0.0498*** | -0.00576 | 0.0731*** | 0.0200 | 0.0172 | 0.0118 | 0.0201 | 0.0297* |
| Parental income | | | | | | | | | | | | | | |
| Medium | -0.0134 | 0.00740 | 0.00417 | 0.0314** | 0.00505 | 0.000162 | -0.00262 | -0.0163 | -0.00231 | -0.00899 | 0.00692 | 0.0171 | 0.00263 | -0.00342 |
| High | 0.0140 | 0.0464*** | 0.0384*** | 0.0448*** | 0.0189 | 0.0504*** | -0.00591 | 0.00149 | -0.00173 | 0.000641 | -0.00292 | 0.0323** | 0.0218 | 1.62e-05 |
| Missing | 0.0561 | 0.0260 | 0.0848 | 0.0628 | 0.0549 | 0.0843 | 0.0521 | 0.206** | 0.0205 | -0.00503 | -0.0191 | -0.0802 | -0.127 | -0.0515 |
| Gender | | | | | | | | | | | | | | |
| Girl | -0.0402*** | -0.0596*** | 0.0248*** | -0.0431*** | 0.00963 | 0.0457*** | 0.0615*** | 0.0308*** | 0.0329*** | -0.0286*** | 0.0165** | 0.0401*** | 0.0624*** | 0.102*** |
| Migration background | -0.0156 | -0.0992*** | 0.0140 | 0.0463*** | 0.0207 | 0.140*** | 0.0682*** | 0.0751*** | -0.0650*** | 0.0818*** | 0.0578*** | 0.0827*** | 0.0494*** | 0.0490*** |
| Migration background | | | | | | | | | | | | | | |
| Additional controls | X | X | X | X | X | X | X | X | X | X | X | X | X | X |
| Constant | 0.0382** | 0.0155 | -0.0562*** | 0.0302 | 0.0562*** | -0.0293 | -0.0173 | 0.0108 | -0.0308 | 0.0331* | -0.00197 | -0.0172 | 0.00465 | 0.0246 |
| Observations | 54,937 | 68,272 | 68,347 | 49,158 | 62,643 | 66,810 | 71,630 | 70,610 | 47,764 | 70,385 | 70,668 | 75,043 | 79,239 | 69,534 |
| R-squared | 0.002 | 0.010 | 0.005 | 0.011 | 0.011 | 0.011 | 0.007 | 0.003 | 0.011 | 0.005 | 0.004 | 0.006 | 0.014 | 0.027 |
| Clusters | 1038 | 1154 | 1149 | 1088 | 1148 | 1161 | 1158 | 1153 | 1076 | 1159 | 1167 | 1166 | 1171 | 1159 |

| VARIABLES | Reading | Spelling | Math |
|---|---|---|---|
| COVID-19 year (2019/2020) | -0.138*** | -0.145*** | -0.206*** |
| Parental education | | | |
| Medium | 0.00815 | 0.00854 | 0.0101 |
| High | 0.0730*** | 0.0408*** | 0.0308*** |
| Missing | 0.0172* | 0.0272*** | 0.0208** |
| Parental income | | | |
| Medium | 0.00700 | -0.00485 | 0.00283 |
| High | 0.0378*** | 0.0131* | 0.0121* |
| Missing | 0.0654 | 0.111** | -0.0565 |
| Gender | | | |
| Girl | -0.0283*** | 0.0361*** | 0.0384*** |
| Migration background | | | |
| Migration background | -0.0164*** | 0.0553*** | 0.0635*** |
| Constant | -0.00110 | -0.00158 | 0.00603 |
| Observations | 240,714 | 319,457 | 364,869 |
| R-squared | 0.005 | 0.007 | 0.009 |

(Continued)

**Table 11.** (Continued)

| VARIABLES | Reading | | | | Spelling | | | | | Math | | | | |
|---|---|---|---|---|---|---|---|---|---|---|---|---|---|---|
| | Grade 2 | Grade 3 | Grade 4 | Grade 5 | Grade 1 | Grade 2 | Grade 3 | Grade 4 | Grade 5 | Grade 1 | Grade 2 | Grade 3 | Grade 4 | Grade 5 |
| Clusters | 1174 | | | | 1172 | | | | | 1178 | | | | |

The outcome variable, learning gain between the midterm and the end-of-term test, has been standardized within grade using the inverse probability population weighted means and standard deviations. The baseline category for "COVID-19 year" are the pooled students from the 2017/2018 and 2018/2019 school years. The baseline category for gender is "boy". The baseline category for parental education is "low". Additional controls include student gender, students' migration background, parental income and a dummy indicating whether students took their end-of-term test at the start of the next, rather than at the end of the current school year. Observations are weighted using entropy weights. Standard errors are clustered at the school level and are omitted for brevity.

* $p < 0.10$; ** $p < 0.05$; *** $p < 0.01$.

**Table 12. Underlying regression results parental education (Fig 2).**

| VARIABLES | Reading | | | | Spelling | | | | | Math | | | | |
|---|---|---|---|---|---|---|---|---|---|---|---|---|---|---|
| | Grade 2 | Grade 3 | Grade 4 | Grade 5 | Grade 1 | Grade 2 | Grade 3 | Grade 4 | Grade 5 | Grade 1 | Grade 2 | Grade 3 | Grade 4 | Grade 5 |
| COVID-19 year (2019/2020) | -0.0583 | -0.135*** | -0.114*** | -0.178*** | -0.268*** | -0.199*** | -0.191*** | -0.155*** | -0.196*** | -0.153*** | -0.266*** | -0.263*** | -0.349*** | -0.371*** |
| Parental education | | | | | | | | | | | | | | |
| Medium | -0.00281 | 0.0410** | 0.0291* | 0.0147 | -0.0244 | -0.0226 | 0.00785 | -0.0211 | 0.0770*** | 0.0251 | -0.0208 | -0.0275 | -0.00181 | 0.0146 |
| High | 0.0224 | 0.108*** | 0.0982*** | 0.0413** | -0.0548** | 0.0173 | 0.0205 | -0.0199 | 0.114*** | -0.00595 | -0.0236 | -0.0108 | -0.00206 | 0.0140 |
| Missing | -0.0373 | 0.0396* | 0.0267 | 0.0458* | -0.0178 | 0.00225 | 0.0411* | -0.0182 | 0.0568** | 0.0151 | -0.0343 | -0.0294 | 0.00399 | 0.0181 |
| COVID-19 year * Medium educated parent | -0.0401 | -0.0529 | -0.0446 | -0.0333 | 0.0725** | 0.0513 | -0.00978 | 0.0443 | -0.0354 | -0.00573 | 0.0906*** | 0.0645** | 0.0329 | 0.0327 |
| COVID-19 year * High educated parent | 0.00703 | -0.0200 | 0.0205 | -0.00591 | 0.139*** | 0.103*** | 0.0754*** | 0.0806** | 0.0436 | 0.0451 | 0.185*** | 0.128*** | 0.138*** | 0.0640* |
| COVID-19 year * Education parent missing | 0.0422 | -0.0283 | 0.00670 | -0.0689 | 0.108** | 0.0226 | 0.0274 | 0.0385 | 0.0517 | 0.0154 | 0.167*** | 0.134*** | 0.0494 | 0.0357 |
| Parental income | | | | | | | | | | | | | | |
| Medium | -0.0130 | 0.00730 | 0.00386 | 0.0309** | 0.00463 | -0.000659 | -0.00272 | -0.0169 | -0.00250 | -0.00923 | 0.00641 | 0.0173 | 0.00153 | -0.00367 |
| High | 0.0148 | 0.0463*** | 0.0382*** | 0.0447*** | 0.0193 | 0.0502*** | -0.00542 | 0.00136 | -0.00129 | 0.000943 | -0.00218 | 0.0331** | 0.0215 | 0.000132 |
| Missing | 0.0553 | 0.0274 | 0.0840 | 0.0605 | 0.0560 | 0.0882 | 0.0548 | 0.206** | 0.0220 | -0.00480 | -0.0159 | -0.0815 | -0.128 | -0.0517 |
| Additional controls | x | x | x | x | x | x | x | x | x | x | x | x | x | x |
| Constant | 0.0368* | 0.00677 | -0.0567*** | 0.0234 | 0.0879*** | -0.00757 | -0.00552 | 0.0291 | -0.0250 | 0.0398** | 0.0411** | 0.0131 | 0.0316* | 0.0391** |
| Observations | 54,937 | 68,272 | 68,347 | 49,158 | 62,643 | 66,810 | 71,630 | 70,610 | 47,764 | 70,385 | 70,668 | 75,043 | 79,239 | 69,534 |
| R-squared | 0.002 | 0.010 | 0.005 | 0.011 | 0.011 | 0.011 | 0.008 | 0.003 | 0.011 | 0.005 | 0.005 | 0.007 | 0.015 | 0.027 |
| Clusters | 1038 | 1154 | 1149 | 1088 | 1148 | 1161 | 1158 | 1153 | 1076 | 1159 | 1167 | 1166 | 1171 | 1159 |

| VARIABLES | Reading | Spelling | Math |
|---|---|---|---|
| COVID-19 year (2019/2020) | -0.127*** | -0.201*** | -0.287*** |
| Parental education | | | |
| Medium | 0.0217** | 0.000913 | -0.00391 |
| High | 0.0713*** | 0.0129 | -0.00727 |
| Missing | 0.0212* | 0.0122 | -0.00268 |
| COVID-19 year * Medium educated parent | -0.0409** | 0.0266 | 0.0470*** |
| COVID-19 year * High educated parent | 0.00430 | 0.0870*** | 0.119*** |
| COVID-19 year * Education parent missing | -0.0126 | 0.0476** | 0.0746*** |
| Parental income | | | |
| Medium | 0.00684 | -0.00520 | 0.00235 |
| High | 0.0379*** | 0.0133* | 0.0123* |
| Missing | 0.0658 | 0.112** | -0.0558 |
| Additional controls | X | X | X |
| Constant | -0.00451 | 0.0160 | 0.0314*** |
| Observations | 240,714 | 319,457 | 364,869 |

*(Continued)*

**Table 12.** (Continued)

| VARIABLES | Reading | | | | Spelling | | | | | Math | | | | |
|---|---|---|---|---|---|---|---|---|---|---|---|---|---|---|
| | Grade 2 | Grade 3 | Grade 4 | Grade 5 | Grade 1 | Grade 2 | Grade 3 | Grade 4 | Grade 5 | Grade 1 | Grade 2 | Grade 3 | Grade 4 | Grade 5 |
| R-squared | 0.005 | | | | 0.007 | | | | | 0.009 | | | | |
| Clusters | 1174 | | | | 1172 | | | | | 1178 | | | | |

The outcome variable, learning gain between the midterm and the end-of-term test, has been standardized within grade using the inverse probability population weighted means and standard deviations. The baseline category for "COVID-19 year" are the pooled students from the 2017/2018 and 2018/2019 school years. The baseline category for household income is "low". The baseline category for parental education is "low". Additional controls include student gender, students' migration background, and a dummy indicating whether students took their end-of-term test at the start of the next, rather than at the end of the current school year. Observations are weighted using entropy weights. Standard errors are clustered at the school level and are omitted for brevity.

* $p < 0.10$; ** $p < 0.05$; *** $p < 0.01$.

**Table 13. Underlying regression results household income (Fig 3).**

| VARIABLES | Reading | | | | Spelling | | | | | Math | | | | |
|---|---|---|---|---|---|---|---|---|---|---|---|---|---|---|
| | Grade 2 | Grade 3 | Grade 4 | Grade 5 | Grade 1 | Grade 2 | Grade 3 | Grade 4 | Grade 5 | Grade 1 | Grade 2 | Grade 3 | Grade 4 | Grade 5 |
| COVID-19 year (2019/2020) | -0.0710* | -0.163*** | -0.127*** | -0.207*** | -0.279*** | -0.225*** | -0.210*** | -0.155*** | -0.200*** | -0.168*** | -0.309*** | -0.298*** | -0.372*** | -0.388*** |
| Parental income | | | | | | | | | | | | | | |
| Medium | -0.000276 | 0.0458*** | 0.0315* | 0.0200 | -0.0227 | -0.0174 | 0.0118 | -0.0206 | 0.0775*** | 0.0291* | -0.0126 | -0.0214 | 0.00193 | 0.0175 |
| High | 0.0235 | 0.116*** | 0.102*** | 0.0532** | -0.0514*** | 0.0314 | 0.0311* | -0.0141 | 0.113*** | 0.00332 | -0.00498 | 0.00297 | 0.00420 | 0.0174 |
| Missing | -0.0348 | 0.0463** | 0.0297 | 0.0546** | -0.0150 | 0.0106 | 0.0471** | -0.0165 | 0.0567** | 0.0197 | -0.0217 | -0.0198 | 0.00942 | 0.0214 |
| COVID-19 year * Medium income | 0.0356 | 0.0637*** | 0.0306 | 0.0595* | 0.0224 | 0.0585** | 0.0403 | -0.00504 | 0.0101 | 0.0396 | 0.0980*** | 0.0752*** | 0.0507** | 0.0397 |
| COVID-19 year * High household income | -0.00810 | 0.0447 | 0.0213 | 0.0836** | 0.0237 | 0.105*** | 0.0756** | 0.0516* | -0.0138 | 0.0622** | 0.129*** | 0.0958*** | 0.0397 | 0.0174 |
| COVID-19 year * Household income missing | 0.0142 | 0.103 | -0.194 | 0.292 | 0.283 | -0.0361 | -0.304 | -0.0853 | -0.168 | -0.242 | 0.244 | 0.211 | 0.0658 | -0.296 |
| Parental education | | | | | | | | | | | | | | |
| Medium | -0.000276 | 0.0458*** | 0.0315* | 0.0200 | -0.0227 | -0.0174 | 0.0118 | -0.0206 | 0.0775*** | 0.0291* | -0.0126 | -0.0214 | 0.00193 | 0.0175 |
| High | 0.0235 | 0.116*** | 0.102*** | 0.0532** | -0.0514*** | 0.0314 | 0.0311* | -0.0141 | 0.113*** | 0.00332 | -0.00498 | 0.00297 | 0.00420 | 0.0174 |
| Missing | -0.0348 | 0.0463** | 0.0297 | 0.0546** | -0.0150 | 0.0106 | 0.0471** | -0.0165 | 0.0567** | 0.0197 | -0.0217 | -0.0198 | 0.00942 | 0.0214 |
| COVID-19 year * Medium educated parent | 0.0356 | 0.0637*** | 0.0306 | 0.0595* | 0.0224 | 0.0585** | 0.0403 | -0.00504 | 0.0101 | 0.0396 | 0.0980*** | 0.0752*** | 0.0507** | 0.0397 |
| COVID-19 year * High educated parent | -0.00810 | 0.0447 | 0.0213 | 0.0836** | 0.0237 | 0.105*** | 0.0756** | 0.0516* | -0.0138 | 0.0622** | 0.129*** | 0.0958*** | 0.0397 | 0.0174 |
| COVID-19 year * Education parent missing | 0.0142 | 0.103 | -0.194 | 0.292 | 0.283 | -0.0361 | -0.304 | -0.0853 | -0.168 | -0.242 | 0.244 | 0.211 | 0.0658 | -0.296 |
| Additional controls | X | X | X | X | X | X | X | X | X | X | X | X | X | X |
| Constant | 0.0412** | 0.0160 | -0.0524*** | 0.0334 | 0.0915*** | 0.00150 | 0.000365 | 0.0290 | -0.0239 | 0.0449** | 0.0559*** | 0.0247 | 0.0393** | 0.0449** |
| Observations | 54,937 | 68,272 | 68,347 | 49,158 | 62,643 | 66,810 | 71,630 | 70,610 | 47,764 | 70,385 | 70,668 | 75,043 | 79,239 | 69,534 |
| R-squared | 0.002 | 0.010 | 0.005 | 0.011 | 0.011 | 0.011 | 0.008 | 0.004 | 0.011 | 0.005 | 0.005 | 0.007 | 0.015 | 0.027 |
| Clusters | 1038 | 1154 | 1149 | 1088 | 1148 | 1161 | 1158 | 1153 | 1076 | 1159 | 1167 | 1166 | 1171 | 1159 |

| VARIABLES | Reading | Spelling | Math |
|---|---|---|---|
| COVID-19 year (2019/2020) | -0.146*** | -0.214*** | -0.313*** |
| Parental income | | | |
| Medium | -0.00825 | -0.0141** | -0.0174*** |
| High | 0.0280*** | -0.00545 | -0.00898 |
| Missing | 0.0540 | 0.126** | -0.0620 |
| COVID-19 year * Medium household income | 0.0450*** | 0.0269** | 0.0593*** |
| COVID-19 year * High household income | 0.0292* | 0.0578*** | 0.0647*** |
| COVID-19 year * Missing household income | 0.0346 | -0.0420 | 0.0170 |
| Parental education | | | |
| Medium | 0.0251*** | 0.00352 | 0.000974 |

(Continued)

**Table 13.** (Continued)

| VARIABLES | Reading | | | | Spelling | | | | | Math | | | | |
|---|---|---|---|---|---|---|---|---|---|---|---|---|---|---|
| | Grade 2 | Grade 3 | Grade 4 | Grade 5 | Grade 1 | Grade 2 | Grade 3 | Grade 4 | Grade 5 | Grade 1 | Grade 2 | Grade 3 | Grade 4 | Grade 5 |
| High | 0.0765*** | | | | 0.0206** | | | | | 0.00255 | | | | |
| Missing | 0.0259** | | | | 0.0165 | | | | | 0.00442 | | | | |
| COVID-19 year * Medium educated parent | -0.0511*** | | | | 0.0185 | | | | | 0.0320** | | | | |
| COVID-19 year * High educated parent | -0.0114 | | | | 0.0637*** | | | | | 0.0885*** | | | | |
| COVID-19 year * Education parent missing | -0.0253 | | | | 0.0355* | | | | | 0.0552*** | | | | |
| Additional controls | X | | | | X | | | | | X | | | | |
| Constant | 0.00198 | | | | 0.0203* | | | | | 0.0403*** | | | | |
| Observations | 240,714 | | | | 319,457 | | | | | 364,869 | | | | |
| R-squared | 0.005 | | | | 0.007 | | | | | 0.009 | | | | |
| Clusters | 1174 | | | | 1172 | | | | | 1178 | | | | |

The outcome variable, learning gain between the midterm and the end-of-termtest, has been standardized within grade using the inverse probability population weighted means and standard deviations. The baseline category for "COVID-19 year" are the pooled students from the 2017/2018 and 2018/2019 school years. The baseline category for household income is "low". The baseline category for parental education is "low". Additional controls include student gender, students' migration background, and a dummy indicating whether students took their end-of-term test at the start of the next, rather than at the end of the current school year. Observations are weighted using entropy weights. Standard errors are clustered at the school level and are omitted for brevity.

* p<0.10; ** p<0.05; *** p<0.01.

**Table 14. Underlying regression results migration background (Fig 4).**

| VARIABLES | Reading | | | | Spelling | | | | | Math | | | | |
|---|---|---|---|---|---|---|---|---|---|---|---|---|---|---|
| | Grade 2 | Grade 3 | Grade 4 | Grade 5 | Grade 1 | Grade 2 | Grade 3 | Grade 4 | Grade 5 | Grade 1 | Grade 2 | Grade 3 | Grade 4 | Grade 5 |
| COVID-19 year (2019/2020) | -0.0300 | -0.114*** | -0.101*** | -0.172*** | -0.253*** | -0.199*** | -0.185*** | -0.157*** | -0.179*** | -0.141*** | -0.239*** | -0.230*** | -0.341*** | -0.360*** |
| Migration background | 0.00233 | -0.0855*** | 0.0233* | 0.0509*** | 0.0311** | 0.141*** | 0.0717*** | 0.0735*** | -0.0535*** | 0.0899*** | 0.0758*** | 0.105*** | 0.0554*** | 0.0566*** |
| COVID-19 year * Migration background | -0.0566* | -0.0425* | -0.0285 | -0.0128 | -0.0322 | -0.000294 | -0.0108 | 0.00477 | -0.0364 | -0.0248 | -0.0559** | -0.0688*** | -0.0177 | -0.0235 |
| Parental education | | | | | | | | | | | | | | |
| Medium | 0.00253 | 0.0446*** | 0.0317* | 0.0159 | -0.0213 | -0.0226 | 0.00879 | -0.0215 | 0.0804*** | 0.0274* | -0.0156 | -0.0216 | -0.000165 | 0.0168 |
| High | 0.0296 | 0.113*** | 0.102*** | 0.0429** | -0.0508*** | 0.0173 | 0.0218 | -0.0205 | 0.119*** | -0.00287 | -0.0166 | -0.00249 | 0.000222 | 0.0170 |
| Missing | -0.0318 | 0.0436** | 0.0297 | 0.0472** | -0.0153 | 0.00228 | 0.0421** | -0.0187 | 0.0610*** | 0.0170 | -0.0292 | -0.0231 | 0.00589 | 0.0208 |
| COVID-19 year * Medium educated parent | -0.0573 | -0.0650* | -0.0520* | -0.0368 | 0.0629* | 0.0512 | -0.0129 | 0.0455 | -0.0457 | -0.0130 | 0.0743** | 0.0445 | 0.0281 | 0.0261 |
| COVID-19 year * High educated parent | -0.0154 | -0.0364 | 0.00998 | -0.0107 | 0.127*** | 0.103*** | 0.0712*** | 0.0824** | 0.0294 | 0.0357 | 0.164*** | 0.101*** | 0.132*** | 0.0552 |
| COVID-19 year * Education parent missing | 0.0287 | -0.0394 | -0.000753 | -0.0727 | 0.103** | 0.0226 | 0.0245 | 0.0398 | 0.0400 | 0.0110 | 0.154*** | 0.116*** | 0.0445 | 0.0286 |
| Additional controls | X | X | X | X | X | X | X | X | X | X | X | X | X | X |
| Constant | -0.00949 | 0.000330 | -0.0612*** | 0.0214 | 0.0829*** | -0.00762 | -0.00717 | 0.0298 | -0.0308 | 0.0360* | 0.0323 | 0.00275 | 0.0288 | 0.0353* |
| Observations | 54,937 | 68,272 | 68,347 | 49,158 | 62,643 | 66,810 | 71,630 | 70,610 | 47,764 | 70,385 | 70,668 | 75,043 | 79,239 | 69,534 |
| R-squared | 0.002 | 0.010 | 0.005 | 0.011 | 0.011 | 0.011 | 0.008 | 0.003 | 0.011 | 0.005 | 0.005 | 0.007 | 0.015 | 0.027 |
| Clusters | 1038 | 1154 | 1149 | 1088 | 1148 | 1161 | 1158 | 1153 | 1076 | 1159 | 1167 | 1166 | 1171 | 1159 |

| VARIABLES | Reading | Spelling | Math |
|---|---|---|---|
| COVID-19 year (2019/2020) | -0.111*** | -0.193*** | -0.268*** |
| Migration background | -0.00588 | 0.0607*** | 0.0762*** |
| COVID-19 year * Migration background | -0.0320** | -0.0162 | -0.0387*** |
| Parental education | | | |
| Medium | 0.0246*** | 0.00240 | -0.000317 |
| High | 0.0753*** | 0.0149 | -0.00239 |
| Missing | 0.0244** | 0.0137 | 0.00115 |
| COVID-19 year * Medium educated parent | -0.0498*** | 0.0220 | 0.0359** |
| COVID-19 year * High educated parent | -0.00783 | 0.0808*** | 0.104*** |
| COVID-19 year * Education parent missing | -0.0212 | 0.0435** | 0.0647*** |
| Additional controls | X | X | X |
| Constant | -0.00949 | 0.0135 | 0.0254** |
| Observations | 240,714 | 319,457 | 364,869 |
| R-squared | 0.005 | 0.007 | 0.009 |

(*Continued*)

**Table 14.** (Continued)

| VARIABLES | Reading | | | | Spelling | | | | | Math | | | | |
|---|---|---|---|---|---|---|---|---|---|---|---|---|---|---|
| | Grade 2 | Grade 3 | Grade 4 | Grade 5 | Grade 1 | Grade 2 | Grade 3 | Grade 4 | Grade 5 | Grade 1 | Grade 2 | Grade 3 | Grade 4 | Grade 5 |
| Clusters | 1174 | | | | 1172 | | | | | 1178 | | | | |

Note: the outcome variable, learning gain between the midterm and the end-of-term test, has been standardized within grade using the inverse probability population weighted means and standard deviations. The baseline category for "COVID-19 year" are the pooled students from the 2017/2018 and 2018/2019 school years. The baseline category for migration background is "no migration background". The baseline category for parental education is "low". Additional controls include student gender, students' migration background, parental income and a dummy indicating whether students took their end-of-term test at the start of the next, rather than at the end of the current school year. Observations are weighted using entropy weights. Standard errors are clustered at the school level and are omitted for brevity.

\* $p < 0.10$; \*\* $p < 0.05$; \*\*\* $p < 0.01$.

**Table 15. Underlying regression results gender.**

|  | Reading | | | | Spelling | | | | | Math | | | | |
|---|---|---|---|---|---|---|---|---|---|---|---|---|---|---|
| VARIABLES | Grade 2 | Grade 3 | Grade 4 | Grade 5 | Grade 1 | Grade 2 | Grade 3 | Grade 4 | Grade 5 | Grade 1 | Grade 2 | Grade 3 | Grade 4 | Grade 5 |
| COVID-19 year (2019/2020) | -0.0399 | -0.118*** | -0.104*** | -0.160*** | -0.275*** | -0.212*** | -0.186*** | -0.162*** | -0.190*** | -0.131*** | -0.263*** | -0.249*** | -0.336*** | -0.370*** |
| Gender |  |  |  |  |  |  |  |  |  |  |  |  |  |  |
| Girl | -0.0278*** | -0.0490*** | 0.0311*** | -0.0314** | 0.00547 | 0.0370*** | 0.0640*** | 0.0263*** | 0.0373*** | -0.0139 | 0.0177* | 0.0490*** | 0.0704*** | 0.102*** |
| COVID-19 year * Girl | -0.0380* | -0.0326* | -0.0195 | -0.0349 | 0.0137 | 0.0261 | -0.00852 | 0.0134 | -0.0128 | -0.0448** | -0.00501 | -0.0281 | -0.0252 | -0.00222 |
| Parental education |  |  |  |  |  |  |  |  |  |  |  |  |  |  |
| Medium | -0.00257 | 0.0411** | 0.0292* | 0.0147 | -0.0243 | -0.0228 | 0.00791 | -0.0211 | 0.0770*** | 0.0250 | -0.0208 | -0.0274 | -0.00170 | 0.0146 |
| High | 0.0227 | 0.108*** | 0.0984*** | 0.0415** | -0.0548*** | 0.0171 | 0.0206 | -0.0200 | 0.114*** | -0.00605 | -0.0236 | -0.0106 | -0.00181 | 0.0140 |
| Missing | -0.0369 | 0.0400** | 0.0269 | 0.0460* | -0.0178 | 0.00203 | 0.0412* | -0.0183 | 0.0569** | 0.0150 | -0.0343 | -0.0291 | 0.00425 | 0.0181 |
| COVID-19 year * Medium educated parent | -0.0397 | -0.0535 | -0.0448 | -0.0337 | 0.0726** | 0.0512 | -0.00997 | 0.0444 | -0.0355 | -0.00591 | 0.0906*** | 0.0641** | 0.0324 | 0.0327 |
| COVID-19 year * High educated parent | 0.00746 | -0.0208 | 0.0199 | -0.00698 | 0.139*** | 0.103*** | 0.0752** | 0.0810*** | 0.0432 | 0.0448 | 0.185*** | 0.127*** | 0.138*** | 0.0639* |
| COVID-19 year * Education parent missing | 0.0424 | -0.0291 | 0.00597 | -0.0697 | 0.108** | 0.0225 | 0.0272 | 0.0389 | 0.0512 | 0.0154 | 0.167*** | 0.133*** | 0.0484 | 0.0356 |
| Additional controls | X | X | X | X | X | X | X | X | X | X | X | X | X | X |
| Constant | 0.0302 | 0.00121 | -0.0600*** | 0.0174 | 0.0901*** | -0.00313 | -0.00699 | 0.0313 | -0.0272 | 0.0327** | 0.0402** | 0.00840 | 0.0273 | 0.0387** |
| Observations | 54,937 | 68,272 | 68,347 | 49,158 | 62,643 | 66,810 | 71,630 | 70,610 | 47,764 | 70,385 | 70,668 | 75,043 | 79,239 | 69,534 |
| R-squared | 0.002 | 0.010 | 0.005 | 0.011 | 0.011 | 0.011 | 0.008 | 0.003 | 0.011 | 0.005 | 0.005 | 0.007 | 0.015 | 0.027 |
| Clusters | 1038 | 1154 | 1149 | 1088 | 1148 | 1161 | 1158 | 1153 | 1076 | 1159 | 1167 | 1166 | 1171 | 1159 |

| VARIABLES | Reading | Spelling | Math |
|---|---|---|---|
| COVID-19 year (2019/2020) | -0.111*** | -0.205*** | -0.273*** |
| Gender |  |  |  |
| Girl | -0.0185*** | 0.0335*** | 0.0472*** |
| COVID-19 year * Girl | -0.0299*** | 0.00771 | -0.0272*** |
| Parental education |  |  |  |
| Medium | 0.0218** | 0.000886 | -0.00385 |
| High | 0.0716*** | 0.0128 | -0.00713 |
| Missing | 0.0215** | 0.0121 | -0.00252 |
| COVID-19 year * Medium educated parent | -0.0413** | 0.0267 | 0.0468*** |
| COVID-19 year * High educated parent | 0.00369 | 0.0872*** | 0.118*** |
| COVID-19 year * Education parent missing | -0.0133 | 0.0478** | 0.0741*** |
| Additional controls | X | X | X |
| Constant | -0.00972 | 0.0173 | 0.0269** |
| Observations | 240,714 | 319,457 | 364,869 |
| R-squared | 0.005 | 0.007 | 0.009 |

(Continued)

**Table 15.** (Continued)

| VARIABLES | Reading | | | | | Spelling | | | | | Math | | | | |
|---|---|---|---|---|---|---|---|---|---|---|---|---|---|---|---|
| | Grade 2 | Grade 3 | Grade 4 | Grade 5 | | Grade 1 | Grade 2 | Grade 3 | Grade 4 | Grade 5 | Grade 1 | Grade 2 | Grade 3 | Grade 4 | Grade 5 |
| Clusters | 1174 | | | | | 1172 | | | | | 1178 | | | | |

Note: the outcome variable, learning gain between the midterm and the end-of-term test, has been standardized within grade using the inverse probability population weighted means and standard deviations. The baseline category for "COVID-19 year" are the pooled students from the 2017/2018 and 2018/2019 school years. The baseline category for gender is "boy". The baseline category for parental education is "low". Additional controls include student gender, students' migration background, parental income and a dummy indicating whether students took their end-of-term test at the start of the next, rather than at the end of the current school year. Observations are weighted using entropy weights. Standard errors are clustered at the school level and are omitted for brevity.

* $p < 0.10$; ** $p < 0.05$; *** $p < 0.01$.

income to better resources these students had: Students with higher-educated parents most likely all possessed a laptop, had parents that were able and willing to help with schoolwork and could even afford additional private tutoring if needed.

The results call for national focus on reducing the learning loss of students from lower-educated parents and lower household income. It is worrisome and unfortunately not unlikely that the increased inequalities in learning loss due to the pandemic may lead to long lasting inequalities, deepening the gap in adult outcomes between groups in the population. This very much stresses the need for targeted interventions to reduce the current inequalities in learning loss caused by the pandemic.

This article shows that schools matter, specifically for the most vulnerable groups. Distance learning may prevent part of the damage but cannot compensate for classroom teaching. The policy implications of these findings are therefore twofold: 1) Government budgets that are made available to make up for learning loss should give schools with many students from low-educated parents and low household income a large share of the pie, and 2) in an event of another crisis, or the current COVID-19-pandemic continues, schools should be closed only as a very last resort to avoid further inequalities.

## Author Contributions

**Conceptualization:** Carla Haelermans, Roxanne Korthals, Stan Vermeulen, Tijana Prokic-Breuer, Rolf van der Velden, Sanne van Wetten, Inge de Wolf.

**Data curation:** Carla Haelermans, Roxanne Korthals, Madelon Jacobs, Suzanne de Leeuw, Stan Vermeulen, Lynn van Vugt.

**Formal analysis:** Carla Haelermans, Roxanne Korthals, Madelon Jacobs, Suzanne de Leeuw, Stan Vermeulen, Lynn van Vugt.

**Funding acquisition:** Carla Haelermans, Roxanne Korthals, Tijana Prokic-Breuer, Rolf van der Velden, Inge de Wolf.

**Investigation:** Carla Haelermans, Roxanne Korthals.

**Methodology:** Carla Haelermans, Roxanne Korthals, Madelon Jacobs, Suzanne de Leeuw, Stan Vermeulen, Lynn van Vugt.

**Project administration:** Carla Haelermans, Roxanne Korthals, Tijana Prokic-Breuer, Rolf van der Velden, Inge de Wolf.

**Supervision:** Tijana Prokic-Breuer, Rolf van der Velden, Inge de Wolf.

**Validation:** Carla Haelermans, Roxanne Korthals.

**Visualization:** Madelon Jacobs, Suzanne de Leeuw, Stan Vermeulen.

**Writing – original draft:** Carla Haelermans, Roxanne Korthals, Madelon Jacobs, Suzanne de Leeuw, Stan Vermeulen, Lynn van Vugt.

**Writing – review & editing:** Carla Haelermans, Roxanne Korthals, Bas Aarts, Tijana Prokic-Breuer, Rolf van der Velden, Sanne van Wetten, Inge de Wolf.

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
