## [Decision Letter · Decision Letter 0]

20 Aug 2021

PONE-D-21-21410

Sharp increase in inequality in education in times of the COVID-19-pandemic

PLOS ONE

Dear Dr. Haelermans,

Thank you for submitting your manuscript to PLOS ONE. After careful consideration, we feel that it has merit but does not fully meet PLOS ONE’s publication criteria as it currently stands. Therefore, we invite you to submit a revised version of the manuscript that addresses the points raised during the review process.

We look forward to receiving your revised manuscript.

Kind regards,

Gabriel A. Picone

Academic Editor

PLOS ONE

Journal Requirements:

2. In the Methods section and the online submission form, please provide additional information about the participant records used in your study. Specifically, please ensure that you have discussed whether all data were fully anonymized before you accessed them.

[The authors gratefully acknowledge financial support from the Dutch Research Council (NWO) and The Netherlands Organisation for Health Research and Development (ZonMw) (project 10430 03201 0014). ]

 [The authors gratefully acknowledge financial support from the Dutch Research Council (NWO) and The Netherlands Organisation for Health Research and Development (ZonMw, https://www.zonmw.nl/nl/) (project 10430 03201 0014). 

Grant acquired by CH, RK, SV, TP-B, RvdV, SvW, IdW]

Reviewers' comments:

Reviewer's Responses to Questions

**Comments to the Author**

1. Is the manuscript technically sound, and do the data support the conclusions?

Reviewer #1: Yes

2. Has the statistical analysis been performed appropriately and rigorously? 

Reviewer #1: Yes

3. Have the authors made all data underlying the findings in their manuscript fully available?

Reviewer #1: Yes

4. Is the manuscript presented in an intelligible fashion and written in standard English?

Reviewer #1: Yes

5. Review Comments to the Author

Reviewer #1: In general, the article looks interesting and gives good and useful information. There are, however, some minor points that must be corrected before publication.

- One of the most important arguments of this article, compared to previous similar publications is:

"we look in greater detail at background differences between students and present results showing that the learning loss due

to the school closures are unequally distributed and that students from disadvantaged backgrounds have suffered much more than their fellow students."

I would like to understand better what is the meaning of "greater detail" since a new publication should offer something different and it should be clearly explained.

- Getting data from 2018 regarding internet access in Dutch households seems to be a source of information that can be updated for sure with more recent data sources.

- Materials and methods: Standardized tests had the same format in the pandemic? Were they online or face-to-face?

I think authors should (if possible) to describe a little bit better the correction CITO made related to the delay in the tests. This fact is very important because authors are analyzing differences in those tests, actually. We must be sure that there is not any influence in that sense.

- Another point that can be improved is the selection of groups when analyzing parents' conditions. It could be done with easy clustering algorithms to check if their groups are correctly separated. Alternatively, a better explanation about why authors selected three groups (instead of four or two, for example) could be adequate.

-Typos: "the school closures and the COVID-1919-pandemic than others"

"paper comes down to 201819 students in 1178"

"This is implies that during the"

...

Congratulations for your nice work.

6. PLOS authors have the option to publish the peer review history of their article (what does this mean?). If published, this will include your full peer review and any attached files.

Reviewer #1: **Yes: **G. M. Sacha

---

## [Author Response · Author response to Decision Letter 0]

7 Sep 2021

Sharp increase in inequality in education in times of the COVID-19-pandemic

PLOS ONE

September 2021

First, let us express our gratitude for your and the reviewers’ constructive comments and remarks. They have helped us a lot in revising and essentially improving the paper. We believe that we have been able to address all of your comments. We hope that you will be satisfied by the respective revisions and replies. Below is a point-by-point response to the main comments that you stressed in the decision email.

Response to main points mentioned by the editor

- Thank you for reminding us about this and our apologies that we did not have this before. We have now adjusted the manuscript to the journals style requirements, including the referencing style.

In the Methods section and the online submission form, please provide additional information about the participant records used in your study. Specifically, please ensure that you have discussed whether all data were fully anonymized before you accessed them.

- We have now added the following to the manuscript on page 7:

“Note that the data in the environment of Statistics Netherlands are pseudonymized such that data are fully anonymous to the researchers that use these data. The pseudonymization key is only known to Statistics Netherlands and they provide separate datasets with the same person identifier, such that the data can be matched, but individuals cannot be identified by the researchers.”

Please remove any funding-related text from the manuscript and let us know how you would like to update your Funding Statement.

- Thank you for pointing out that we did not do this in the correct way. We have now removed the funding related text from the manuscript. The Funding statement already was and still is complete, so that does not need to be changed. 

In your Data Availability statement, you have not specified where the minimal data set underlying the results described in your manuscript can be found. PLOS defines a study's minimal data set as the underlying data used to reach the conclusions drawn in the manuscript and any additional data required to replicate the reported study findings in their entirety.

- Our apologies, it seems that something went wrong with the data availability statement. We have adjusted that now. Please let us know should the current data availability statement not be sufficient. 

Please review your reference list to ensure that it is complete and correct.

- Thank you. We have now updated our reference list, and made sure that it’s complete and correct.

Response to reviewer 1

First, let us express our gratitude for your positive words, and constructive comments and remarks. They have helped us a lot in revising and essentially improving the paper. We believe that we have been able to address all of your and their comments. We hope that you will be satisfied by the respective revisions and replies.

In general, the article looks interesting and gives good and useful information. 

- Thank you.

There are, however, some minor points that must be corrected before publication.

One of the most important arguments of this article, compared to previous similar publications is:

"we look in greater detail at background differences between students and present results showing that the learning loss due to the school closures are unequally distributed and that students from disadvantaged backgrounds have suffered much more than their fellow students."

I would like to understand better what is the meaning of "greater detail" since a new publication should offer something different and it should be clearly explained.

- Thank you for this comment. We should indeed have been more specific in discussing our contribution to the existing literature with this study. We have now added some additional explanation to the manuscript, and the text on page 3 we have now added the following:

“In comparison with the study that is most similar that also uses Dutch data (Engzell et al. 2020), our study complements and improves the findings for several reasons. 1) We have a larger sample (18% of all Dutch primary schools, instead of 15%), 2) we have much better and richer student background information at the individual level (instead of socio-economic status measured at the neighborhood level, which is very imprecise), 3) we have multiple student background variables that indicate whether a student is disadvantaged or not, and 4) we focus on effects for separate grade levels, showing large variation there, instead of only looking at overall effects.”

Getting data from 2018 regarding internet access in Dutch households seems to be a source of information that can be updated for sure with more recent data sources.

- Thank you for pointing this out. You are right, and we have now replaced the source by the most recently available data, which is from 2020. The share of households with internet access is the same though, so we did not change the percentage in the text.

Materials and methods: Standardized tests had the same format in the pandemic? Were they online or face-to-face?

- Yes indeed, the standardized test during the pandemic were exactly the same as before the pandemic. For most schools, these are digital tests that take place at school. Some schools opt for the pen-and-paper version. But there is there is no within school variation between type and format of the tests before and during the pandemic. We have now also added this information to the manuscript on page 6. 

I think authors should (if possible) to describe a little bit better the correction CITO made related to the delay in the tests. This fact is very important because authors are analyzing differences in those tests, actually. We must be sure that there is not any influence in that sense.

- You are raising a fair point here. Unfortunately, the details of the correction that CITO has applied to these tests are only known to CITO, and not shared with schools or researchers using these data. However, CITO have a long history of excellent expertise in test development, calibration and correction, so we trust in CITOs many years of experience in these matters.

Having said that, we did include a dummy in all our analyses whether a student participated in a test before or after the summer break, and this does not influence our results. Furthermore, we have also checked whether we find similar results when we only focus on the students that wrote the test before the summer break (so in the regular time frame of the test), and we do not find different results or draw different conclusions. Therefore, we are not worried that the corrected test influences the results in any way. 

Another point that can be improved is the selection of groups when analyzing parents' conditions. It could be done with easy clustering algorithms to check if their groups are correctly separated. Alternatively, a better explanation about why authors selected three groups (instead of four or two, for example) could be adequate.

- Thank you for pointing out that we should have explained this in a better way. We have now done that in both the description of the student background variables, and in the description of the results. 

On page 7 we have added the following sentence:

“This division of parental education over three categories is also being used in the Netherlands Cohort Study on Education and leads to a division in categories that is not only relevant at the content level, but also provides us with large enough groups to have statistical power.”

Furthermore, on page 15 we have added the following sentence to the results:

“(Note that alternative specifications in which we use four categories of parental education (in a similar way as the Dutch Inspectorate of Education), or in which we use three categories which are not based on parental education, but on the indication (used for funding purposes) whether a child is a regular child, has a disadvantaged background or a very disadvantaged background, yield very similar results and the same conclusions.)”

Typos: "the school closures and the COVID-1919-pandemic than others"

"paper comes down to 201819 students in 1178"

"This is implies that during the"

- Thank you, we have corrected the typos in the text and checked once more for other typos as well. 

Congratulations for your nice work.

- Thank you, and thank you once more for your valuable comments. We hope we have been able to deal with your comments in a satisfactory way.

---

## [Decision Letter · Decision Letter 1]

19 Oct 2021

PONE-D-21-21410R1Sharp increase in inequality in education in times of the COVID-19-pandemicPLOS ONE

Dear Dr. Haelermans, Let me first apologize for the delay in processing your manuscript on behalf of PLOS ONE. I took over the editorial duty and, because I always aim to obtain  two expert opinions on every manuscript, decided to ask for an additional expertise. You will find the review at the bottom of this email. As you will see, this second reviewer is quite positive about your study but asks for some clarifications. I concur with the reviewer that your study is timely, interesting and relevant and I would encourage you to address the reviewer's comment.

We look forward to receiving your revised manuscript.

Kind regards,

Jérôme Prado

Academic Editor

PLOS ONE

Reviewers' comments:

Reviewer #2: This an interesting and very timely study on pertinent issues in education policy. Below are some concerns I have about the study. Clarifying those would be helpful.

Major Comments:

1. How does the sample used in the study compare to the overall school population in the Netherlands? In other words, how representative is the sample of Netherlands nationally? The authors refer to using a national sample in calculating weights. How do averages of various variables in the study sample compare to those in the national sample?

2. Relatedly, what proportion of total schools in Netherlands does the sample of schools, students and test records in Table 2 represent?

3. On page 6 authors mention that “Test supplier CITO made a recalculation for all test scores to correct for this delayed testing.” Is CITO the only supplier that did this recalculation or did other mentioned suppliers did it too? What kind of impact can we expect this calculation to have on the scores, especially in comparison to other suppliers and to previous years scores? While authors cannot get to the actual calculation CITO did, some explanation would be helpful.

4. Do the authors have any information on teachers or school level variables? Does the study account for teacher and school level effects in their estimations? For example we can expect some teachers to do better in remote instruction that others. Are/Can the estimations accounting for this in any way?

5. Parental Education: They mention on page 7 that “Parental education is defined as low

when the highest obtained degree of (one) the parents is in pre-vocati…..”. They mention further down in the text that they use the higher of the two parents educational attainment. Would it not be pertinent instead to also look at this my mother/father or by the parent with the larger share of child responsibilities? Perhaps, if we expect the parent with lower educational attainment to be more responsible for child caring, look at estimates along that margin?

6. The school closure periods mentioned on page 4 is not very long. How do authors see this in light of closure in other parts of the world e.g. U.S. where schools remained closed for extended periods of time? Can we expect more widening of gaps across different groups if closure remained longer?

7. During the time period of the closure, the vulnerable kids were apparently still allowed to attend schools. How does this, if at all, interact with the income and education of the households. Are authors able to identify students who continued attending during the closure?

8. Results: Education literature generally finds that educational interventions bring a larger change in math and a smaller change in reading scores, partly because reading is not just dependent on what is taught in school but requires stronger input from home also. It would be nice to tie in that literature with the Math vs. Reading losses the authors estimates

9. How should we see migrants in terms of income and education? In other words what is the average education level of migrants in Netherlands and what income category should we expect them to fall into. In other words, more clarity on needed for the reader on analysis over migration vs. income or education.

10. It is not clear what Table 3 is showing. Is it the number of test score observations? If yes, why do we have decimals? If no, then are these some averages of test scores ?

11. Figure 5, 6 and 7 need to be clearer. I could not understand what was being shown by each line.

Minor Comments

1. Relevant study to cite on the effect of pandemic on student learning : https://gpl.gsu.edu/download/student-achievement-growth-during-the-covid-19-pandemic-report-appendix/?wpdmdl=2101&refresh=614b7200638111632334336

2. Contribution: I think in terms of contribution the authors need to think beyond the Dutch data. They emphasize on page 3 the comparison to other work that uses Dutch data. I would urge them to look at other studies, in different countries, that look at inequality in education outcomes during the pandemic and situate their study in the wider literature.

3. The authors mention on page 3 their contribution compared to an existing study on Dutch data. I don’t think having an 18% vs 15% sample is a contribution, unless the new sample is more representative for some reason. The other points about having better and richer background info is certainly something to point out.

Typos

Page 25: “figs” should be replaced by Figures

Page 4 : “ At total of 96% of Dutch households “have” internet acces…”

Page 2: For higher education the results are less consistent: some find negative effects [6] while others indicate that distance learning might have made students more efficient [7] or see little effects [8].

---

## [Author Response · Author response to Decision Letter 1]

15 Nov 2021

Sharp increase in inequality in education in times of the COVID-19-pandemic

PLOS ONE

November 2021

Response to reviewer 2

First, let us express our gratitude for your positive words, and constructive and insightful comments and remarks. They have helped us a lot in revising and essentially improving the paper. We believe that we have been able to address all of your and their comments. Based on the comments we extended our discussion of the international literature on the role of the COVID-19 pandemic in children’s educational development, we performed additional robustness checks (e.g., school fixed effects) and clarified some of our tables and figures. We address the comments of the reviewer hereafter in more detail in a point-by-point response. We hope that you will be satisfied by the respective revisions and replies.

This an interesting and very timely study on pertinent issues in education policy.

Thank you.

Below are some concerns I have about the study. Clarifying those would be helpful.

1. How does the sample used in the study compare to the overall school population in the Netherlands? In other words, how representative is the sample of Netherlands nationally? The authors refer to using a national sample in calculating weights. How do averages of various variables in the study sample compare to those in the national sample?

Schools are only included in our sample if they gave permission to share their test scores with Statistics Netherlands. As a result, our sample is not an exact representation of the population of Dutch primary school students. We added Table 3 to the manuscript to show how our study sample compares to the full population. Overall, our sample contains an overrepresentation of one-parent households (19.00% versus 16.16% in the full population), students with a non-western migration background (20.82% versus 14.36%), students with low parental income (24.06% versus 21.20%), and an underrepresentation of students from which both parents work (68.47% versus 70.91%). Furthermore, schools in our sample tend to be larger schools located in more urbanized areas. As you mention already, we use inverse probability weights to minimize the impact of the selectivity of our sample. 

 

Table 3. Representativeness of sample compared to full population on student and school background variables 

 Full population Sample

Variables Percentage Percentage

Gender 

Female 49.32 49.75

Migration background 

Dutch & western migration background 82.22 75.95

Non-western migration background 17.51 24.04

Missing 0.27 0.01

Parental income 

Low income 21.20 24.06

Medium income 53.38 50.54

High income 24.02 24.53

Missing 1.41 0.87

Parental education 

Low educated 10.06 11.50

Medium educated 29.87 29.03

High educated 47.59 48.95

Missing 12.48 10.52

School size 

Less than 141 students 36.71 28.95

Between 141 – 220 students 30.27 31.26

More than 220 students 33.01 39.79

School level pct of low educated parents 

Below 5,5% 33.12 32.22

Between 5,5% and 12% 33.47 32.78

Above 12% 33.41 35.00

Urbanisation level 

Low (< 500 adresses/km2) 7.86 5.63

Limited (500 – 1000 adresses/km2) 21.87 14.41

Medium (1000 – 1500 adresses/km2 17.58 12.50

Strong (1500 – 2500 adresses/km2) 30.93 32.66

Very strong (>=2500 adresses/km2) 21.77 34.81

Denomination 

Public school 29.79 30.55

Schools based on philosophies 6.10 3.77

Schools based on religious beliefs 64.02 65.69

Observations 

Total number 2,458,376 263,553

Note: N of students is based on the number of unique students in the years 2017/2018, 2018/2019 and 2019/2020

2. Relatedly, what proportion of total schools in Netherlands does the sample of schools, students and test records in Table 2 represent?

Thank you for pointing out that this was not clear. In the schoolyear 2019/2020, we had a total number of 6174 primary schools in the Netherlands. The 1178 schools in our sample therefore comprise a proportion of 19% of the total number of schools. 

Having said that, we realised that the distinction between test supplier and administrative system could be confusing (see your next question and our answer). Moreover, the distinction between the different administration systems is not relevant for our analyses. Therefore, we decided to no longer mention the names of the administration systems in our paper and to remove Table 2.

3. On page 6 authors mention that “Test supplier CITO made a recalculation for all test scores to correct for this delayed testing.” Is CITO the only supplier that did this recalculation or did other mentioned suppliers did it too? What kind of impact can we expect this calculation to have on the scores, especially in comparison to other suppliers and to previous years scores? While authors cannot get to the actual calculation CITO did, some explanation would be helpful.

Again: thank you for making us realise this was unclear. Based on your question we realised that there might be confusion between what a test supplier is and what an administrative system is. In the previous version of the paper, we made a distinction between test suppliers and administrative systems. All children in our sample made tests of a supplier called CITO. CITO is by far the largest test supplier in the Netherlands. The test scores of the CITO-tests are saved in an administration system. Schools can choose themselves which administration system they prefer. Our dataset is based on the data stored in three types of administration systems: CITO-LOVS, ParnasSys and ESIS. Former table 2 showed the number of schools per administration system.

However, based on your comment, we realised that the names CITO-LOVS (administration system) and CITO (test supplier) could confuse our readers. As explained in our answer to your previous question, we have now removed Table 2.

So as an answer to your questions 3: there is only one test supplier that we include in our analyses, namely CITO. Therefore, the scores of all students in our sample who performed the test after the summer were corrected. By correcting for the delay, the test supplier aimed to make the test scores comparable to the test scores of students who took the test before summer, to account for the extra time the students had until the test was taken. Note that the test supplier did not correct for the fact that this test was taken in the period of COVID-19 in any way, meaning that they did not correct for lower scores compared to previous years. To make sure our results are not biased by the differences in the timing of the test we also include a dummy in our regression analyses indicating whether a test was made before (=0) or after (=1) summer. Based on your comment, we improved our explanation of the differences in test scores before and after the summer on page 6.

4. Do the authors have any information on teachers or school level variables? Does the study account for teacher and school level effects in their estimations? For example we can expect some teachers to do better in remote instruction that others. Are/Can the estimations account for this in any way?

Thank you for this question. We do have information on school level variables, but unfortunately we do not have any information on the teachers. We agree that school and teacher level factors play a role in the extent to which the COVID-19 related school closings affected student learning. The association between student characteristics and learning gains might be driven by, for example, differences between schools serving different types of student populations. To account for this possibility, we have added a new analysis incorporating Fixed Effects at the school level. These can be found in Table 9 (split by grade level) on page 24 and Table 10 (pooled over all grades) on page 25. In this specification, we account for any (un)observed time invariant differences between schools that could have an effect on the strength of the relationship between student characteristics and the COVID-19 related reductions in learning gains. The results show that the estimations remain unaffected. The possibility that different types of teachers may differentially affect student learning gains unfortunately cannot be tested, as we do not have information on the teachers to whom students have been assigned. 

5. Parental Education: They mention on page 7 that “Parental education is defined as low

when the highest obtained degree of (one) the parents is in pre-vocati…..”. They mention further down in the text that they use the higher of the two parents educational attainment. Would it not be pertinent instead to also look at this my mother/father or by the parent with the larger share of child responsibilities? Perhaps, if we expect the parent with lower educational attainment to be more responsible for child caring, look at estimates along that margin?

Thank you for this very relevant question. However, we believe that in our setup, parents’ highest obtained level of education is still the best proxy for children’s socioeconomic background for several reasons. One is that we cannot be certain as to which parent has the larger share of child responsibilities. Especially during the time of the pandemic and the increase in remote working, the balance between work and family life is likely to have shifted compared to previous years. This might render interpretation of the results difficult. Secondly, the administrative data on parents’ highest obtained level of education does not yet fully cover the entire population of the Netherlands. In combining the education data from both mother and father, the share of missing observations is reduced. Due to assortative mating there is a relatively high correlation between the highest obtained level of education of both mother and father. Hence, the education of one of the parents is also an indicator of the educational attainment of the other parent. 

Nevertheless, you are right that it is instructive to look at the sensitivity of our results to running estimations including either the mother’s or father’s education, or both at the same time, instead of the highest level. The results of this exercise show that pooled over all grades, the main association between parental education and learning loss during the pandemic holds for all three subjects. For reading, maternal education enters more significantly than paternal education, while for the other subjects there is no difference. In the analyses by grade, we again see no differences for spelling and math. For reading, high paternal education appears to be more significantly related to learning losses in grades 3, 4, and 5. However, these results do not survive our standard robustness specifications. Despite the small differences in results based on paternal and maternal education and children’s reading, our overall conclusions do not change when paternal education or maternal education is used. Because of these results, as well as the aforementioned conceptual reasons, we strongly favour the models in which the highest level of parental education is used. 

6. The school closure periods mentioned on page 4 is not very long. How do authors see this in light of closure in other parts of the world e.g. U.S. where schools remained closed for extended periods of time? Can we expect more widening of gaps across different groups if closure remained longer?

We agree with you that children in countries with longer school closings and less internet access might have experienced larger learning losses and larger inequalities because they experienced prolonged periods of limited and unequal excess to education. However, the current literature is inconclusive and more research is needed to draw pertinent conclusions on the association between the length of the lockdown and children’s educational progress. For example, a recent study in Italy by Contini et al (2021) reports larger learning losses (0.19 SD) than previous studies in the Netherlands (0.08 SD) by Engzell et al (2021). In Italy schools were closed for 15 weeks (one of the first and longest school closings in Europe). Moreover, Italy has one of the lowest share of households with a broadband connection and 12% of the students between 6 and 17 years old did not have access to a computer or digital tools at home in 2018/2019. However, contradicting the idea that longer school closings result in larger learning deficits, a study in Belgium (8.5 weeks of school closing) reports a reduction in mathematics scores of 0.19 SD which is similar in size to the effects found in Italy (15 weeks). Altogether, more research based on country comparisons is needed to be able to state that longer lockdowns result in larger learning deficits and an increase in educational inequalities. We have added a paragraph to reflect on country differences on page 4 and 5.

7. During the time period of the closure, the vulnerable kids were apparently still allowed to attend schools. How does this, if at all, interact with the income and education of the households. Are authors able to identify students who continued attending during the closure?

Thank you for this question. When the Dutch government decided to close schools, some children still went to school. The government prescribed that schools could allow ‘vulnerable’ children as well as children of parents with essential occupations at school. These rules on which children could and could not attend school were purposely vague and schools could mostly define their own policies. As a result, we do not know which children went to school and which children stayed at home. However, especially during the first lockdown, which is the subject of our study, schools were only open for emergencies and the number of children at school was very low. Based on a survey by the General Association of School Leaders about 5% of the students went to school during the first lockdown. Moreover, children who went to school during the school closings usually followed a program which was similar to the program of the children who stayed at home. We have now also added a few sentences on this on page 4. Finally, due to the various reasons to let children spend their day at school, the group of children at school was diverse and exceeded low socioeconomic status families. Since the number of children that attended school is not registered in our data, we cannot make a distinction between children who attended school and who did not attend school during the school closure so we cannot examine the differences in performances based on school attendance.

Whether the presence of a small group of children at school is problematic for the analyses depends on your goal. We argue that we estimate the effect of the Dutch educational policies during the pandemic on children’s education progress. Allowing a selective group of children at school is part of the educational policies and therefore not problematic for our analyses. Nevertheless, we acknowledge that school attendance by vulnerable groups might cause a bias if one wants to estimate the effect of distance education on educational progress. If a bias would occur, one might argue that the results reported in our studies underestimate the inequalities based on parents’ education and income since some inequalities might have been cushioned by school attendance. However, we argue that the bias would be small since the number of children allowed at school was small and the group of children relatively diverse. Moreover, the educational program of children at school and children at home was relatively similar. 

8. Results: Education literature generally finds that educational interventions bring a larger change in math and a smaller change in reading scores, partly because reading is not just dependent on what is taught in school but requires stronger input from home also. It would be nice to tie in that literature with the Math vs. Reading losses the authors estimates

Thank you for the helpful literature suggestion. The idea that family environment plays an important role in the development of reading skills is supported by the finding that reading skills are less affected by the school closure. The relative importance of schools might be smaller for reading than for other subjects which explains the limited impact of the pandemic. In contrast, the limited increase in socioeconomic inequalities in reading skills is not in line with our expectations. Normally, family environment is an important source of inequality in reading skills and we would expect inequalities to rise when schools closed and the role of family environment increased. We have added a discussion on the differences between reading and mathematics before and during the pandemic in the Conclusions on page 32 and 33.

9. How should we see migrants in terms of income and education? In other words what is the average education level of migrants in Netherlands and what income category should we expect them to fall into. In other words, more clarity on needed for the reader on analysis over migration vs. income or education.

You are right that we should clarify to the international audience how students with a non-western migrant background should be seen in terms of parental education and household income. We have added the comparison of the share of low educated parents and low household income between students with a non-western migration background and native / western migrant students on page 8. The share of students with low educated parents and a low household income is higher for those with a non-western migration background than for those with a native / western migration background (26% vs. 6% for low parental education, 45% vs. 16% for low household income.

10. It is not clear what Table 3 is showing. Is it the number of test score observations? If yes, why do we have decimals? If no, then are these some averages of test scores?

In former Table 3 (Table 2 in the new revised version of the paper) we indeed show the number of observations (test records) per domain and per grade. We accidently used the European way to write this down. In continental Europe the meaning of punctuation marks is the exact opposite from Anglo-Saxon countries. In the Netherlands, and many other European countries, dots are used to separate large numbers while commas are used as decimal markers. We revised our punctuation across the manuscript to the Anglo-Saxon manners with commas to separate large numbers and dots as decimal markers. Hopefully this also clarifies the interpretation of Table 2.

11. Figure 5, 6 and 7 need to be clearer. I could not understand what was being shown by each line.

Thank you for pointing this out. We have first of all made these figures a bit bigger in the manuscript, such that they are better to read. Furthermore, as for explanation what information the figures show: Figures 5, 6 and 7 show the trends in learning gains per grade over time by plotting the unstandardized learning gains by cohort. Because our data does not go back equally far for all grades, the lines are of different length, which perhaps causes some confusion (see also table 1 on page 5). For grade 5, we have data for the grade 5 cohorts of 2017/2018 until 2019/2020, while for grade 2 we have data for students that were in grade 2 from the 2014/2015 academic year onwards. The main message from these figures is that we see a clear drop in learning gains for most grades during the 2019/2020 academic year for the three domains reading, spelling, and math. We have added some additional clarifying sentences regarding these figures on page 27.

Minor Comments 

1. Relevant study to cite on the effect of pandemic on student learning : https://gpl.gsu.edu/download/student-achievement-growth-during-the-covid-19-pandemic-report-appendix/?wpdmdl=2101&refresh=614b7200638111632334336

Thank you for the useful suggestion. We included the report in our discussion of the literature on page 3.

2. Contribution: I think in terms of contribution the authors need to think beyond the Dutch data. They emphasize on page 3 the comparison to other work that uses Dutch data. I would urge them to look at other studies, in different countries, that look at inequality in education outcomes during the pandemic and situate their study in the wider literature.

We have improved the discussion of previous literature in the introduction (page 2 and 3). The new version of the introduction includes more references to the international literature and has been complemented with recent studies which appeared during the review process of our paper. 

3. The authors mention on page 3 their contribution compared to an existing study on Dutch data. I don’t think having an 18% vs 15% sample is a contribution, unless the new sample is more representative for some reason. The other points about having better and richer background info is certainly something to point out.

In our enthusiasm to show the reader that we have a richer and better dataset to examine the consequences of the school closings on educational growth of primary school students in the Netherlands, we argued that our sample is also somewhat larger than the sample used by Engzell et al. Naturally, this is not the strongest argument. As the reviewer already states, our main improvement is based on the rich background information we have in our data. Based on this comment and the previous comment (minor 2) we have rewritten the paragraph on page 3. We now situate our study in the international literature. We emphasize now how the Dutch context offers, due to standardized testing, unique opportunities to study the consequences of the school closings. Moreover, we emphasize that we have, compared to previous studies in the Netherlands, better information on students’ background characteristics. 

Typos

Thank you for pointing them out. We have now corrected the typos.

In addition to your comments, we have also substantially shortened the results section that contained a lot of repetition, and have removed some more typographical errors from the paper. 

Thank you once more for your valuable comments. We hope we have been able to deal with your comments in a satisfactory way.

---

## [Decision Letter · Decision Letter 2]

25 Nov 2021

Sharp increase in inequality in education in times of the COVID-19-pandemic

PONE-D-21-21410R2

Dear Dr. Haelermans,

We’re pleased to inform you that your manuscript has been judged scientifically suitable for publication and will be formally accepted for publication once it meets all outstanding technical requirements.

Kind regards,

Jérôme Prado

Academic Editor

PLOS ONE

Additional Editor Comments (optional):

Reviewers' comments:

Reviewer's Responses to Questions

**Comments to the Author**

1. If the authors have adequately addressed your comments raised in a previous round of review and you feel that this manuscript is now acceptable for publication, you may indicate that here to bypass the “Comments to the Author” section, enter your conflict of interest statement in the “Confidential to Editor” section, and submit your "Accept" recommendation.

Reviewer #2: All comments have been addressed

2. Is the manuscript technically sound, and do the data support the conclusions?

Reviewer #2: (No Response)

3. Has the statistical analysis been performed appropriately and rigorously? 

Reviewer #2: (No Response)

4. Have the authors made all data underlying the findings in their manuscript fully available?

Reviewer #2: (No Response)

5. Is the manuscript presented in an intelligible fashion and written in standard English?

Reviewer #2: (No Response)

6. Review Comments to the Author

Reviewer #2: (No Response)

7. PLOS authors have the option to publish the peer review history of their article (what does this mean?). If published, this will include your full peer review and any attached files.

Reviewer #2: No

---

## [Editor Report · Acceptance letter]

7 Jan 2022

PONE-D-21-21410R2 

Sharp increase in inequality in education in times of the COVID-19-pandemic 

Dear Dr. Haelermans:

I'm pleased to inform you that your manuscript has been deemed suitable for publication in PLOS ONE. Congratulations! Your manuscript is now with our production department. 

Kind regards, 

on behalf of

Dr. Jérôme Prado 

Academic Editor

PLOS ONE